

# Divergent changes in aerosol optical hygroscopicity and new particle

## formation induced by heatwaves

Yuhang Hao[1, a], Peizhao Li[1, a], Yafeng Gou[1], Zhenshuai Wang[1], Mi Tian[1], Yang Chen[2],
Ye Kuang[3], Hanbing Xu[4], Fenglian Wan[1], Yuqian Luo[1], Wei Huang[5], Jing Chen[1, 6, *]

[1] College of Environment and Ecology, Chongqing University, Chongqing 400045,

China

[2] Center for the Atmospheric Environment Research, Chongqing Institute of Green

and Intelligent Technology, Chinese Academy of Sciences, Chongqing 400714, China

[3] Institute for Environmental and Climate Research, Jinan University, Guangzhou

511443, China

[4] Experimental Teaching Center, Sun Yat-sen University, Guangzhou 510275, China

[5] National Meteorological Center, China Meteorological Administration, Beijing

100081, China

[6] Key Laboratory of Three Gorges Reservoir Region's Eco-Environment, Ministry of

Education, Chongqing University, Chongqing 400045, China

[a] These authors contributed equally

*Correspondence to*: Jing Chen (chen.jing@cqu.edu.cn)



**Abstract.** As a crucial climate-forcing driver, the aerosol optical enhancement factor
($f$(RH)) is significantly modulated by the evolution of particle number size
distribution (PNSD), e.g., during new particle formation (NPF). The mechanisms
regulating aerosol optical hygroscopicity during different NPF events and non-event
days, particularly those influenced by heatwaves due to global warming, remain
poorly understood. In the extremely hot summer of 2022 in urban Chongqing of
southwest China, simultaneous measurements of aerosol optical and hygroscopic
properties, PNSD, and bulk chemical compositions were conducted. Two distinct
types of NPF were identified: the ones with relatively polluted period (P1) and clean
cases during heatwave-dominated period (P2). Heatwaves triggered NPF earlier and
prolonged the subsequent growth, resulting in smaller aerosol effective radius ($R_{eff}$)
and lower growth rate. This agreed with the concurrently increased aerosol
hemispheric backscattering fraction and scattering Ångström exponent. $f$(RH) was
generally higher during NPF events in comparison to that for non-event cases in both
periods. Heatwave-induced stronger photooxidation may intensify the formation of
more hygroscopic secondary components, as well as the subsequent growth of
pre-existing particles and newly formed ultrafine ones, thereby enhancing aerosol
optical hygroscopicity especially during heatwave-influenced NPF events. The
promoted $f$(RH) and lowered $R_{eff}$ could synergistically elevate the aerosol direct
radiative forcing, specifically under persistent heatwave conditions. Further in-depth
exploration on molecular-level characterizations and aerosol radiative impacts of both
direct and indirect interactions during weather extremes (e.g., heatwaves) with the
warming climate are recommended.

**1 Introduction**

Weather extremes (e.g., heatwaves) have become more and more frequent and

intense largely due to the global climate change, and the heatwave-driven
environmental, climatic, and health effects have garnered widespread attention



(Hauser et al., 2016; Sun et al., 2016). The China Climate Bulletin 2022 confirmed
that the national average temperature reached an unprecedented high level since 2012
(China Meteorological Administration, 2022), and the risk of heatwaves in China will
persist and potentially intensify in the future (Guo et al., 2016; Li et al., 2017).
Extreme heatwave events could pose significant threats to human health, the survival
of organisms, agriculture, and socio-economic activities (e.g., power supply
restrictions) (Anderson and Bell, 2011; Ma et al., 2021; Su, 2021). Moreover,
heatwaves can trigger natural disasters such as droughts and wildfires, affecting social
stability (Sharma and Mujumdar, 2017).
Heatwaves could also affect the atmospheric physical and chemical processes by
modulating ambient meteorological conditions. Specifically, extremely high
temperature weather is typically characterized by a combination of intensified solar
radiation with elevated temperature and low humidity levels. This could significantly
affect the formation and evolution of secondary aerosols in the atmosphere (Bousiotis
et al., 2021; Hamed et al., 2011; Kurtén et al., 2007), given that the air temperature is
crucial for chemical reactions (Xu et al., 2011). New particle formation (NPF) serves
as a crucial source of atmospheric particulate matter and plays a significant role in the
secondary transformation processes in the atmosphere (Zhu et al., 2021). Generally,
NPF involves the initial formation of thermodynamically stable clusters from
condensable vapors (e.g., ammonia, sulfuric acid, and organic precursor gases) and
subsequent growth of the formed clusters, eventually reaching detectable sizes or even
larger dimensions (Kerminen et al., 2018; Kulmala et al., 2003, 2012). Over time,
these newly formed particles have the potential to serve as cloud condensation nuclei
(CCN), thereby impacting the global climate (Salma et al., 2016). NPF events
normally introduce a sharp increase in the number concentration of nucleation mode
particles within a short time, altering the particle number size distribution (PNSD).
These variations in PNSD likely influence intrinsic physicochemical properties of
aerosols, such as the optical hygroscopicity (Chen et al., 2014; Titos et al., 2016; Zhao
et al., 2019).



Aerosol hygroscopicity plays a critical role in the atmospheric environment and
climate change, given the complex interaction between aerosol particles and water
vapor (Zhao et al., 2019; Zieger et al., 2011). Water uptake by aerosols not only alters
the particle size and composition (e.g., as reflected in the aerosol refractive index) but
also impacts aerosol scattering efficiency, which further contributes to the uncertainty
in aerosol radiative forcing estimation (Titos et al., 2016, 2021). The aerosol optical
hygroscopicity parameter, $f$(RH), defined as the ratio of the scattering coefficient at a
certain RH to that of the dry condition, was widely used to describe the aerosol
scattering enhancement through water uptake (Covert et al., 1972; Titos et al., 2016;
Zhao et al., 2019). Numerous studies have demonstrated that $f$(RH) is influenced by
the size distribution, in addition to particle chemical composition (Chen et al., 2014;
Kuang et al., 2017; Petters and Kreidenweis, 2007; Quinn et al., 2005). NPF could
alter the size distribution thereby aerosol optical properties, nonetheless, there is
currently limited research on the impact of NPF on aerosol optical hygroscopicity (Ma
et al., 2016; Ren et al., 2021). It is suggested that the influence of NPF on aerosol
hygroscopicity was likely due to changes in aerosol chemical composition at different
stages of NPF events (Cheung et al., 2020), whereas the subsequent particle growth
associated with NPF events can significantly affect particle hygroscopicity as well
(Wu et al., 2016). Although there have been a great many studies on chemical
composition dependences of aerosol hygroscopicity (e.g., the variation in composition
of precursor species during NPF events), it is important to acknowledge that the
utilized chemical compositions of NPF were either from $PM_{2.5}$ or $PM_1$ bulk data,
which may differ from the corresponding composition of newly formed ultrafine
particles primarily in the nucleation and Aitken modes. This may further introduce
bias in exploring the impacts of NPF events on aerosol optical hygroscopicity if solely
based on $PM_{2.5}$ chemical composition, especially in the initial nucleation stage of NPF.
Hence, more comprehensive investigations on the influencing mechanisms of aerosol
optical hygroscopicity from different perspectives are required, e.g., for the aspects of
the evolution of particle size distribution in modulating aerosol optical and
hygroscopic properties (Tang et al., 2019; Zhao et al., 2019). Additionally, field





observations on $f$(RH) under extreme weather conditions (e.g., heatwaves) are rather
scarce, largely hindering our understanding of how weather extremes (e.g., extremely
high temperature) influence the optical hygroscopic properties of aerosols. This
knowledge gap further impedes comprehensive understanding of the aerosol water
uptake property and resulted effects on air quality and the climate under varied
synoptic conditions.

During the summer of 2022, a rare heatwave event raged throughout the

Sichuan-Chongqing region of southwest China, with the daily maximum temperature
exceeding 40 ℃ lasted for 29 days observed at Beibei meteorological station in
Chongqing (Hao et al., 2023). This persistent heatwave not only impacted residents'
daily lives significantly, but also affected the aerosol optical and hygroscopic
properties likely through NPF and relevant atmospheric processing during the period.
In this study, a field observation was conducted by using a combination of a
home-built humidified nephelometer system and a scanning mobility particle sizer
(SMPS), along with the total suspended particle (TSP) filter sampling. A main goal of
this study is to investigate the influence of heatwaves on NPF events and subsequent
impacts on aerosol optical and hygroscopic properties. Furthermore, we aimed to
explore the mechanisms behind the variability in $f$(RH) under different meteorological
conditions and NPF events. This study will further enrich insights into the potential
environmental and climatic impacts due to variations in the aerosol optical
hygroscopicity and size distribution, specifically under weather extremes (e.g.,
heatwaves) with the changing climate.

## 2 Data and Methods

### 2.1 Field observation

A continuous field observation on aerosol optical, hygroscopic and chemical

properties was carried out from July 29 to August 19, 2022. The detailed description
of the observation site is available in Supporting Information, S1. During the



observation period, urban Chongqing suffered a rare heatwave (Fig. S1), which
significantly affected the local transportation and industrial activities (Hao et al.,
2023). Based on the temperature records and concurrent aerosol light scattering data,
the whole study period was categorized into two stages: (1) the normally hot period
(with the daily maximum temperature seldomly above 35°C) from 29 July to 6 August
(simply labeled as P1); (2) the heatwave-dominated cleaner period (persistent
occurrence of the hourly temperature over 40°C, and the hourly total scattering
coefficient at 525 nm below 100 Mm$^{-1}$) during 7-19 August 2022 (marked as P2).
**2.2 Instrumentation and methods**
**2.2.1 Measurements of aerosol optical hygroscopicity**
The humidified nephelometer system, consisting of two three-wavelength (i.e.,
450, 525, and 635 nm) nephelometers (Model Aurora 3000, Ecotech Inc.) and a
humidification unit, was used to determine the aerosol light scattering enhancement
factor, $f$(RH). Briefly, the aerosol scattering ($\sigma_{sca, \lambda}$) and backscattering coefficients
($\sigma_{bsca, \lambda}$) were detected in a dry state (RH <30%) and at a fixed RH level of 85% ± 1%,
respectively, with the humidification efficiency regulated automatically by a
temperature-controlled water bath.
Hence, $f$(RH) could be calculated as the ratio of the aerosol scattering coefficient
at a predefined RH ($\sigma_{sca, RH}$) to the dry ($\sigma_{sca, dry}$) state, i.e., $f$(RH) = $\sigma_{sca, RH}$ / $\sigma_{sca, dry}$
(Covert et al., 1972). In this study, the $f$(RH) discussed is mainly targeted for the 525
nm wavelength, unless otherwise specified. More information about the measurement
of humidified nephelometer system was illustrated in S2 of the supplement.
In additional to $f$(RH), aerosol optical parameters, such as scattering Ångström
exponent (SAE; Schuster et al., 2006) and hemispheric backscattering fraction (HBF;
Collaud Coen et al., 2007), were calculated as below:
$$\mathrm{SAE}_{\lambda 1/\lambda 2} = \frac{-\ln\left(\sigma_{sca, \lambda 1}/\sigma_{sca, \lambda 2}\right)}{\ln\left(\lambda 1/\lambda 2\right)} \quad (1)$$

$$\mathrm{HBF}_{\lambda} = \frac{\sigma_{bsca, \lambda}}{\sigma_{sca, \lambda}} \quad (2)$$



where $\sigma_{sca,\,\lambda}$ and $\sigma_{bsca,\,\lambda}$ represent the aerosol scattering and backscattering
coefficients at a specific wavelength $\lambda$ (e.g., $\lambda 1$, $\lambda 2$), respectively.
Both HBF and SAE reflect crucial optical properties of aerosols, e.g., an elevated
HBF (or SAE) generally signifies a higher concentration (or a smaller particle size) of
fine particles within the aerosol population (Jefferson et al., 2017; Kuang et al., 2017;
Luoman et al., 2019). The HBF and SAE discussed in this study are targeted for the
dry condition, unless otherwise specified. Based on the measurements with the
humidified nephelometer system, the equivalent aerosol liquid water content (ALWC)
and the corresponding fraction of ALWC ($f_W$) can also be obtained (Kuang et al, 2018;
see S2 of the supplement).
The SMPS-measured concurrent particle number size distributions were further
utilized to calculate the aerosol effective radius ($R_{eff}$) and representative parameters
for NPF events, e.g., the growth rate (GR) of new particle, condensation sink (CS)
and coagulation sink (CoagS) (Kulmala et al., 2012). More details are provided in the
supplement.

**2.2 Determination of the aerosol direct radiative forcing (ADRF) enhancement**
**factor**
Given the high sensitivity of aerosol optical properties (e.g., $f$(RH)) to the
changes in RH under real atmospheric conditions, the influence of RH, or rather the
aerosol hygroscopicity, on ADRF can be quantitatively estimated with the radiative
transfer model by the following equation (Chylek and Wong, 1995; Kotchenruther et
al., 1999; L. Zhang et al., 2015):
$$\Delta F_R(RH) = -(S_0/4) \times [T_a^2 \times (1 - A_C)] \times [2 \times (1 - R_s)^2 \times \beta(RH) \times \tau_s - 4 \times R_s \times \tau_a] \quad (3)$$
where $S_0$ is the solar constant, $T_a$ is the atmosphere transmittance, $A_C$ is the
fractional cloud amount, $R_s$ is the albedo of the underlying surface, $\beta(RH)$ is the
upscattering fraction at a defined RH, $\tau_s$ and $\tau_a$ are the optical thicknesses of the
aerosol layer due to light scattering and light absorption, respectively, which can be
expressed as follows (Kotchenruther et al., 1999):



$$\tau_s = M \times \alpha_s \times f(RH), \tau_a = M \times \alpha_a \qquad (4)$$
where M is the column burden of aerosol (unit: $gm^{-2}$), $\alpha_s$ is the mass scattering
efficiency (MSE), and $\alpha_a$ is the mass absorption efficiency (MAE). The direct
radiative forcing is usually calculated with the assumption that the absorption
enhancement is negligible, in comparison to the aerosol scattering enhancement (Xia
et al., 2023).
Hence, the dependence of ADRF on RH (i.e., $f_{RF}(RH)$) can be estimated by
equation (5) (Chylek and Wong, 1995; Kotchenruther et al., 1999; L. Zhang et al.,

2015):

$$f_{RF}(RH) = \frac{\Delta F_R(RH)}{\Delta F_R(dry)} = \frac{(1-R_s)^2 \times \beta(RH) \times \alpha_s \times f(RH) - 2 \times R_s \times \alpha_a}{(1-R_s)^2 \times \beta(dry) \times \alpha_s \times f(dry) - 2 \times R_s \times \alpha_a} \qquad (5)$$
where the constant parameters used were $R_s = 0.15$, $\alpha_a = 0.3$ $m^2 \cdot g^{-1}$ (Hand and
Malm, 2007; Fierz-Schmidhauser et al., 2010). It should be noted that the assumed
constant $\alpha_a$ might introduce some uncertainty in the calculated $f_{RF}(RH)$, given the fact
that the contribution of absorption by brown carbon was unknown, although the mass
fraction of BC in TSP remained almost constant (i.e., 4.6% ± 1.1%, Fig. S2) during
the observation period. The parameter $\alpha_s$ was calculated by dividing $\sigma_{sca, 525}$ in the dry
condition by the mass concentration of $PM_{2.5}$ (i.e., $\alpha_s = \sigma_{sca, 525} / PM_{2.5}$). $\beta$ could be
calculated empirically from the measured HBF: $\beta = 0.0817 + 1.8495 \times HBF - 2.9682$
$\times HBF^2$ (Delene and Ogren, 2002).
Results of the offline chemical analysis with TSP filter samples are provided in
S3. Given that the particle number and mass size distributions of components such as
sulfate and organics from diverse emission sources were primarily concentrated
within the submicron size range (An et al., 2024), the bulk chemical compositions of
TSP could provide a reasonably good reference for the characterization of NPF and
related optical and hygroscopic properties of $PM_{2.5}$. It should be noted that the
corresponding mass fraction of some components (e.g., crustal materials) likely
biased for larger particles. The simultaneous meteorological and air quality data can
be found in S4.



## 3 Results and discussion

### 3.1 Overview of the aerosol optical hygroscopicity and PNSD measurements

Figure. 1 displayed the time series of the measured aerosol scattering coefficients, $f(RH)$, PNSD, and the corresponding meteorological conditions and air pollutants during the study period. A sharp decrease in aerosol scattering coefficients and $PM_{2.5}$, accompanied with the continuous excellent visibility over 20 km was observed after August 6, indicating a markedly cleaner environment during P2 in comparison to P1 in summer 2022 of Chongqing. This could be largely attributed to the reduction in anthropogenic emissions (e.g., $NO_2$, CO) from limited outdoor activities influenced by the heatwaves in P2, as well as partly suspended industries and transportation to alleviate the power shortage issue. Notably, the increased wind speed and enhanced mixing layer height (MLH) also enabled a more favorable atmospheric diffusion condition in P2, facilitating the dilution of surface air pollutants (Zhang et al., 2008). However, a higher mass concentration of $SO_2$ was observed in the P2 period, likely due to a surge in electricity demand and resulted higher emissions from power plants operating almost at full capacity during the heatwave (Su, 2021; Teng et al., 2022). Moreover, significant discrepancies in the aerosol optical and hygroscopic properties were observed under different synoptic conditions (Table S2). Both HBF and SAE were higher during the P2 period, aligning with the smaller $R_{eff}$ (Table S2). The $f(RH)$ was found to be larger in heatwave days, with the mean values of 1.6 ± 0.1 and 1.7 ± 0.2 during the P1 and P2 periods, respectively. Differently, ALWC was more abundant during the normally hot P1 period than the heatwave-dominated P2 period, likely due to that the derivation algorithm of ALWC utilized in this study (Kuang et al., 2018) was partly dependent on (e.g., positively correlated) the aerosol scattering coefficient in the dry condition. The mean $\sigma_{sca,\ 525}$ for P2 was about 46.8% of that for the P1 period, and the corresponding mean level of ALWC was approximately 55.8% of that for P1. This partly agrees with the stronger aerosol optical hygroscopicity with a marginally higher $f_W$ during the P2 period, highlighting a complex interaction between the optical enhancement and aerosol physicochemical properties.



The particle number size distribution data suggested that NPF events appeared in
about half the number of observation days (Fig. 1i), with the frequency during the P2
period (53.8%) slightly higher than that of P1 (44.4%). This suggests the rather
frequent summer NPF events in Chongqing, notably higher than those observed in
other regions of the world, e.g., Beijing (16.7%, Deng et al., 2020; ~20%, Wang et al.,
2013), Dongguan (4%, Tao et al., 2023), Hyytiälä (<40%, Dada et al., 2017) and
LiLLE (<20%, Crumeyrolle et al., 2023). Moreover, the frequent NPF events during
heatwaves formed substantially ultrafine particles that are of less contribution to
aerosol optical properties in comparison to large particles, partially explaining the
significantly lower levels of total scattering coefficients observed during the P2
period.



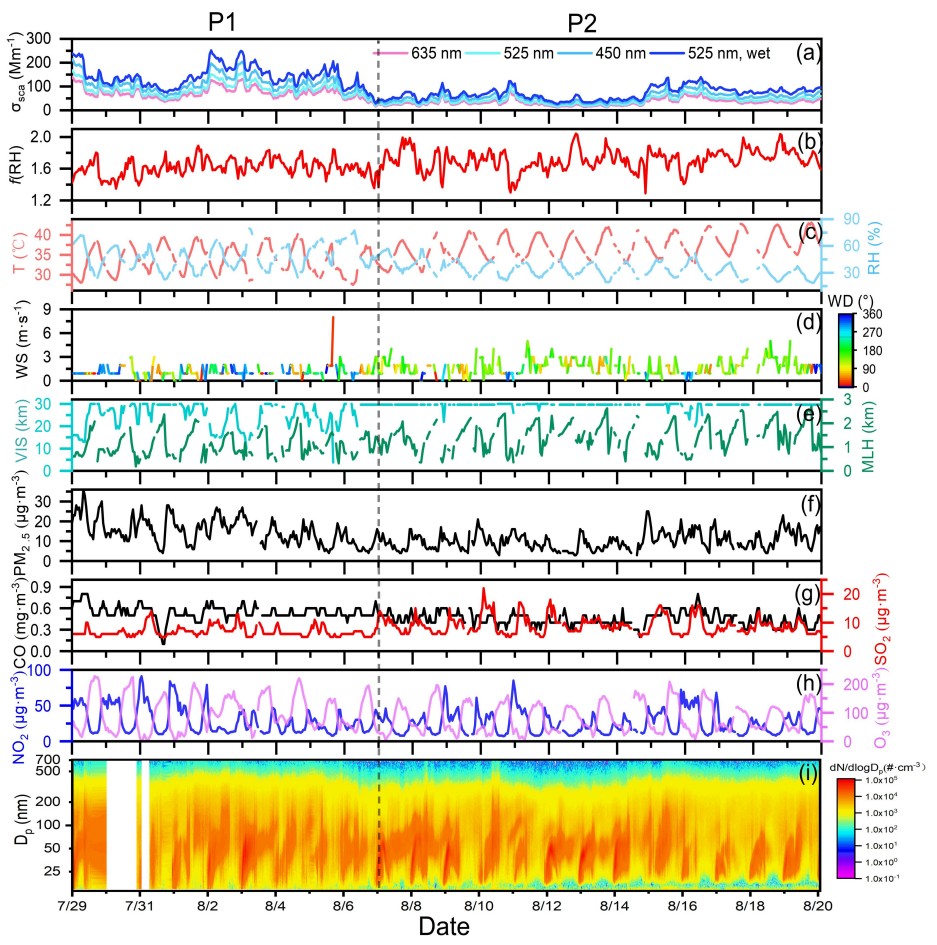

**Figure 1.** Time series of the measured aerosol scattering coefficients, $f$(RH), meteorological conditions, air pollutants, and particle number size distribution during the study period.

### 3.2 Characteristics of NPF events in different periods

Aside from gaseous precursors (e.g., $SO_2$, volatile organic compounds), meteorological conditions also play a key role in the occurrence of NPF events. In brief, NPF events are more likely to appear under sunny and clean conditions (Bousiotis et al., 2021; Crumeyrolle et al., 2023; Deng et al., 2021; Wang et al., 2017). To further explore the characteristics of NPF events in different periods, the



time-averaged diurnal variations of meteorological parameters and air pollutant

concentrations during both NPF events and non-event days are presented in Fig. 2.

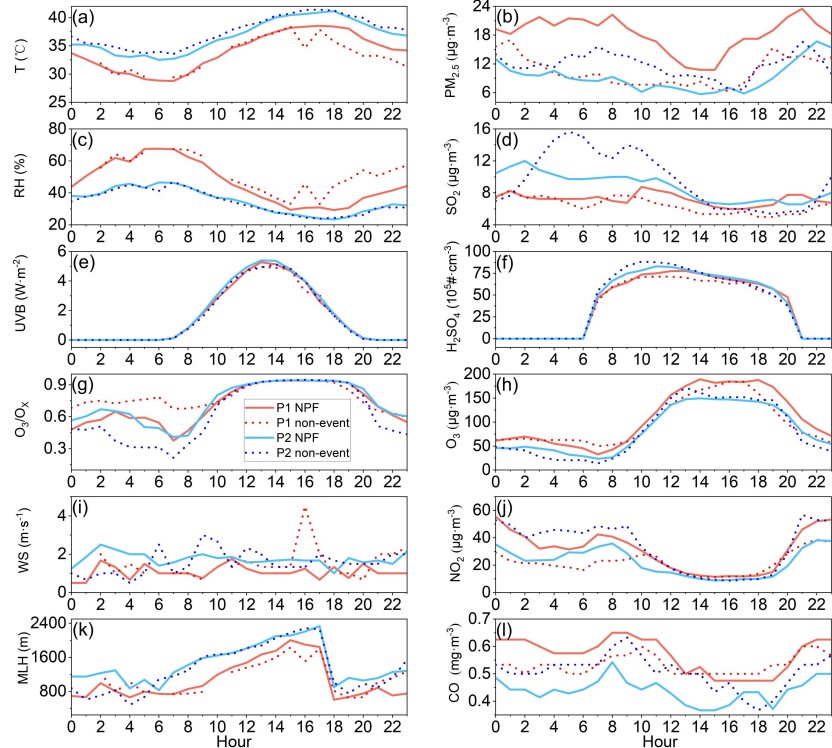

**Figure 2.** Diurnal variations of temperature **(a)**, $PM_{2.5}$ mass loading **(b)**, RH **(c)**, $SO_2$
**(d)**, UVB **(e)**, $H_2SO_4$ **(f)**, $O_3/O_X$ **(g)**, $O_3$ **(h)**, WS **(i)**, $NO_2$ **(j)**, MLH **(k)** and CO **(l)**
during P1 (red) and P2 (blue) NPF events (solid line), as well as the corresponding
non-event days (dash line).

NPF events during the P1 period tended to occur in relatively polluted
environments compared to that of P2 NPF events, as evidenced by the higher $\sigma_{sca,\,525}$,
increased air pollutant concentrations and lower visibility levels during P1 (Table S2,
Fig. 1). On P2 NPF event days, the overall mean $\sigma_{sca,\,525}$ was 33.2 ± 11.7 $Mm^{-1}$,
decreased by 68.0% (39.3%) in comparison to that for P1 NPF event days (P2
non-event days). In addition, the mean $PM_{2.5}$ concentration was even lower than 10.0
$\mu g \cdot m^{-3}$, and the corresponding visibility level was almost reaching the upper detection
limit of 30 km. All the above implies that the P2 NPF events were generally



accompanied with a much cleaner environment. It is notable that the increase in $SO_2$
concentration after 9:00 (Fig. 2d), along with the significant decrease in $PM_{2.5}$ mass
loadings thereby lowered CS or CoagS after 8:00 during P1 NPF events (Fig. 2b),
likely favored the occurrence of NPF events. The higher gas-phase sulfuric acid (i.e.,
$H_2SO_4$, as estimated with the UVB and $SO_2$ concentration, Lu et al., 2019, S4) on the
same NPF event days (Fig. 2f), further suggesting that sulfuric acid concentration was
a critical factor for the occurrence of P1 NPF events.

Meanwhile, the diurnal evolutions of meteorological conditions (e.g., T, RH,

MLH) for NPF events were distinct between P1 and P2 periods, although relatively
insignificant differences were observed for both NPF events and non-event days
within a same period (Fig. 2). This might suggest that meteorological factors might
not be the predominant determining factor of NPF occurrence, while NPF could be
accompanied with quite different meteorological conditions depending on gaseous
precursors and preexisting condensation sinks. For instance, the heatwave-influenced
NPF events were typically of clean-type NPF, characterized with lower background
aerosol loading, higher temperature and favorable atmospheric dispersion capacity
with the higher MLH. However, it is reported that excessive heat can increase the
evaporation rate of critical acid-base clusters during the nucleation process and reduce
the stability of initial molecular clusters (Bousiotis et al., 2021; Kurtén et al., 2007;
Zhang et al., 2012). On the other hand, the emission rate of biogenic VOCs ($BVOC_s$,
e.g., isoprene, monoterpene) from nearby plants and trees would decrease when
temperature exceeded around 40 °C (Guenther et al., 1993; Pierce and Waldruff,
1991), despite that BVOCs plays a key role in the nucleation mechanism of NPF
(Wang et al., 2017; Zhang et al., 2004). Hence, the even higher temperature (e.g.,
T >40 °C) likely hindered the occurrence and subsequent growth of NPF during
non-event days of the P2 period, in spite of higher concentrations of $SO_2$ and $H_2SO_4$.

To further investigate the effect of heatwave on NPF events, the diurnal

variations of aerosol number and volume concentrations, as well as $R_{eff}$, for different
modes were illustrated in Fig. S4, and the relationship between temperature and the
duration of NPF events was displayed in Fig. S5. The NPF events influenced by





heatwaves usually initiated earlier (Fig. S5), with the number concentration of
nucleation mode particles ($N_{Nuc.}$) in P2 NPF cases peaked about an hour earlier (whilst
relatively lower) in comparison to P1 event days (Fig. S4a). This implies that
heatwaves may accelerate the attainment of the temperature threshold of NPF events,
as evidenced by the earlier NPF start time corresponding to higher temperature ranges
(Fig. S5). Furthermore, the end time of subsequent particle growth during P2 period
was even later (i.e., ~ 21:00 LT) than that of P1 cases (Fig. S5). Given the lower GR
during P2 NPF events (Table S2), these explosively formed new particles could
persist longer in the warmer atmosphere and probably undergo aging processes with a
relatively higher oxidation degree. This is supported by the commonly higher ratios of
secondary organic carbon (SOC) to organic carbon (OC) (i.e., SOC/OC >0.5) during
the P2 NPF event days (Fig. S2b). The diurnal patterns of aerosol volume
concentrations for different size modes were similar to that of aerosol number
concentrations during NPF events (Fig. S4b1-b3). It is worth noting that both the $R_{eff}$
of Aitken mode particles ($R_{Ait.}$) and accumulation mode particles ($R_{Acc.}$) were smaller
during P2 NPF events than that of P1 NPF events (Fig. S4c2-c3), which may further
influence size-dependent aerosol optical and hygroscopic properties (e.g., $\sigma_{sca,\ 525}$,
HBF, SAE, $f$(RH)). The decrease in $R_{Ait.}$ and $R_{Acc.}$ during heatwaves could be
attributed to three factors: (1) evaporation of the outer layer of particles due to
extremely high temperature (Bousiotis et al., 2021; Cusack et al., 2013; Deng et al.,
2020; Li et al., 2019); (2) lower GR of particles under a cleaner environment; (3)
reduced emissions of larger primary particles during the P2 period.
**3.3 Characteristics of the aerosol optical and hygroscopic properties during NPF**
**events**
Diurnal variations of the aerosol optical and hygroscopic parameters during NPF
events were shown in Fig. 3, and the corresponding results for non-event days can
refer to Fig. S6. Generally, $\sigma_{sca,\ 525}$ possessed a similar bimodal diurnal pattern to that
of the accumulation mode aerosol volume concentration ($V_{Acc.}$) (Fig. S4b3), as
supported by the positive correlation between $\sigma_{sca,\ 525}$ and SMPS-measured aerosol



volume concentration (Fig. S8). This is also consistent with the Mie theory, with a
stronger increase in the scattering efficiency for accumulation mode particles (Titos et
al., 2021). The diurnal pattern of $\sigma_{sca,\ 525}$ also varied distinctly between different NPF
events. Specifically, a minor peak of $\sigma_{sca,\ 525}$ around 12:00 (Fig. 3a) was influenced by
the newly formed particles during P2 NPF events, which contributed more
significantly to the aerosol number and volume concentrations within 100 nm size
ranges in pretty clean environments (Fig. S3c, g). Instead of a noontime peak, $\sigma_{sca,\ 525}$
was observed with an early peak around the morning rush hours and a maximum
value similarly occurred at the nighttime on P1 NPF event days.

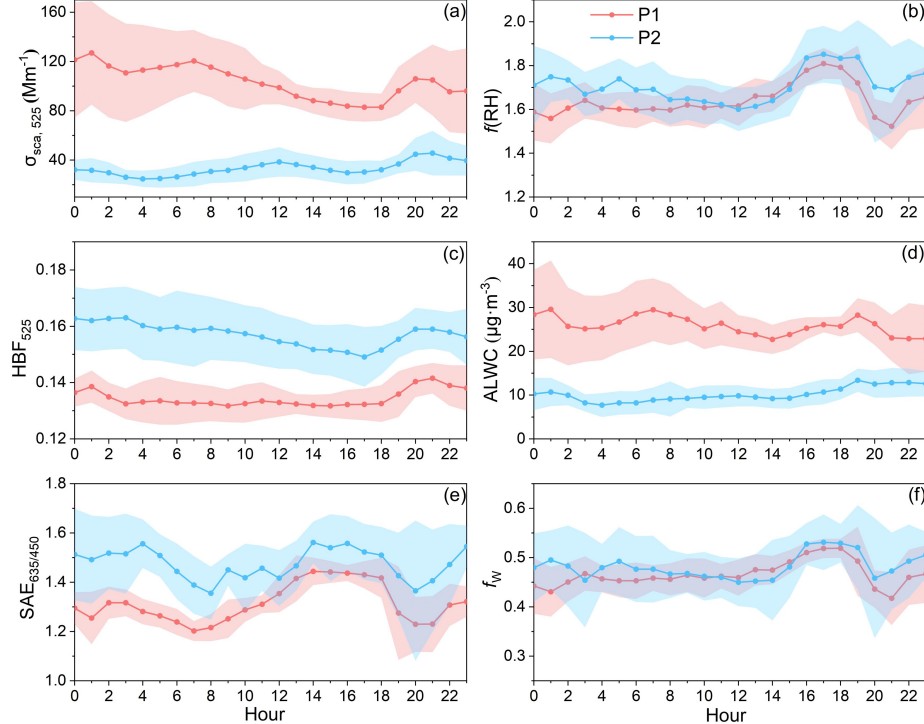


**Figure 3.** Diurnal variations of $\sigma_{sca,\ 525}$ **(a)**, $f$(RH) **(b)**, HBF$_{525}$ **(c)**, ALWC **(d)**,
SAE$_{635/450}$ **(e)** and $f_W$ **(f)** on NPF event days during P1 (red line) and P2 (blue line)
periods. The shaded areas stand for the corresponding ± 1σ standard deviations.
Both HBF and SAE during P2 NPF events were significantly higher than that
of P1 NPF cases (Fig. 3c, e), largely due to the smaller R$_{eff}$ during P2
heatwave-dominated NPF events (Table S2). Moreover, the correlation between HBF



(or SAE) and particle size in each mode was relatively weaker on NPF event days
than on non-event days, especially for P2 NPF events (Fig. S10). A strongest negative
correlation was found between HBF and $R_{eff}$ of the accumulation mode in comparison
to other modes, highlighting that HBF is more sensitive to the size distribution of
accumulation mode particles (Collaud Coen et al., 2007). Given that NPF would
largely enhance the abundance of both nucleation and Aitken mode aerosols, no
significant variation in HBF was observed during the daytime due to the weakened
correlation between HBF and $R_{Acc.}$ of NPF events. SAE is commonly used as an
indicator of particle size distribution, almost decreasing monotonously with the
increase of aerosol size within 1 μm (Kuang et al., 2017, 2018; Luoma et al., 2019).
Accordingly, SAE decreased over the morning and evening rush hours when coarse
particles (e.g., aged particles, road dust, automobile exhaust) generated during
anthropogenic activities, accompanied with an increase in CO that is taken as the
proxy for primary emissions (Fig. 2l) (Yarragunta et al., 2020). On the contrary, the
abundant ultrafine particles formed during NPF events led to a continuous increase in
SAE during the day.
$f$(RH) exhibited a similar diurnal pattern on the P1 and P2 NPF event days
(Fig. 3b). During the daytime, $f$(RH) remained relatively stable and gradually
increased until peaking around 16:00-18:00, with a generally higher $f$(RH)
particularly after 15:00 during P2 NPF events than that of P1 cases. The insignificant
fluctuation of relatively lower $f$(RH) levels before the noon could be attributed to the
continuous development of the mixing layer (Fig. 2k), leading to an efficient mixing
of particles in the nocturnal residual layer with anthropogenic emissions near the
ground. Additionally, photochemical reactions in the afternoon facilitated the
formation of more hygroscopic secondary aerosols with a higher oxidation level (Liu
et al., 2014; R. Zhang et al., 2015). The diurnal patterns of $O_3$ and the $O_3/O_X$ ratio (i.e.,
an indicator of atmospheric oxidation capacity, where $O_X = O_3 + NO_2$, Tian et al.,
2021) also showed similar trends (Fig. 2g, 2h). The presence of black carbon (BC)
mixed with organic compounds (e.g., from traffic emissions and residential cooking
activities) explained the rapid decrease in $f$(RH) during the evening rush hours (Liu et



al., 2011). Furthermore, the daily mean $f$(RH) during NPF events was higher than that
of non-event days (Table S2), particularly after the ending of NPF events around
12:00. Given that newly formed particles were too small to significantly impact the
total light scattering (Fig. S7), this indicates that the atmospheric conditions
conducive to the occurrence of NPF may promote further growth (e.g., via
photooxidation or atmospheric aging processes) of pre-existing particles and newly
formed ones, leading to enhanced aerosol optical hygroscopicity as clued from the
concurrent variations of ALWC and $f_W$ in urban Chongqing during hot summer (Asmi
et al., 2010; Wang et al., 2019; Wu et al., 2016). The diurnal pattern of ALWC closely
mirrored the variation in $\sigma_{sca,\ 525}$, while $f_W$ followed the similar evolution of $f$(RH).
This suggests that ALWC was more sensitive to changes in the aerosol volume
concentration, as determined by the corresponding retrieval algorithm (Kuang et al.,
2018). The relatively higher $f_W$ levels (e.g., even exceeded 50% sometimes) verified
the enhancement of aerosol hygroscopicity during NPF events in comparison to that
of non-event days.

**3.4 Heatwave-induced divergent changes in aerosol optical hygroscopicity**

To further explore the impacts of heatwaves on $f$(RH) during diverse NPF events,
data mainly within the time window of 08:00-22:00 (i.e., typically covered the
complete process of NPF and subsequent growth, while excluded higher RH
conditions at night) were utilized for the following discussion.
A positive correlation between $f$(RH), $R_{eff}$ and the volume fraction of
accumulation mode particles ($VF_{Acc.}$) was found on non-event days (Fig. 4c-d), when
the aerosol size distribution was undisturbed by newly formed ultrafine particles and
the corresponding $VF_{Acc.}$ maintained around a relatively high level of 0.95 (Fig. 4a-b).
The notably positive correlation between $f$(RH) and $R_{eff}$ could be linked to the
secondary formation of hygroscopic particles within the accumulation mode,
primarily via photochemical reactions and further intensified by heatwaves during the
day particularly of the P2 period (Gu et al., 2023; Liu et al., 2014; R. Zhang et al.,
2015; Zhang et al., 2024). Consequently, $f$(RH) at a specific $R_{eff}$ was generally higher





during the P2 period in comparison to that of P1 (Fig. 4c-d), also with high $f$(RH)
levels observed for smaller size cases of $R_{eff}$ <110 nm under some extremely high
temperature conditions (T >40 °C, as highlighted by the red dashed circle in Fig. 4d).
The higher SOC/OC on P2 non-event days further demonstrated the stronger
secondary aerosol formation in comparison to P1 non-event days (Fig. S2b).

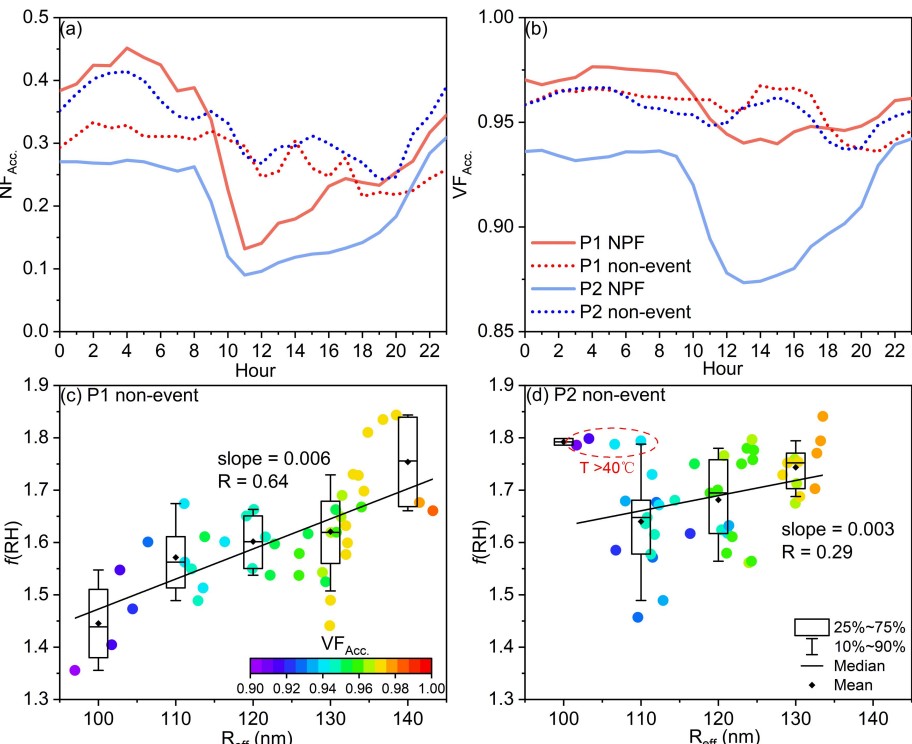


**Figure 4.** Diurnal variations of **(a)** the number fraction (NF$_{Acc.}$) and **(b)** volume

fraction of accumulation mode particles (VF$_{Acc.}$) on P1 (red) and P2 (blue) NPF event
days (solid line), as well as non-event days (dash line). The relationship of $f$(RH) with
$R_{eff}$ and VF$_{Acc.}$ (as indicated by the colored dots) on P1 **(c)** and P2 non-event days **(d)**
during the 08:00-22:00 time window.

Nevertheless, $f$(RH) was almost independent of the two parameters (i.e., $R_{eff}$

and VF$_{Acc.}$) for NPF events (Fig. S11a1-a2). This is mainly due to the explosive
formation of ultrafine particles during NPF events, significantly altering aerosol size
distributions and inducing large fluctuations in the number and volume fractions of



accumulation mode particles (Fig. 4a-b). Therefore, characterizing $f$(RH) with the
corresponding $R_{eff}$ of aerosol populations was no longer applicable. Alternatively,
SAE was commonly used to estimate or parameterize $f$(RH) (Titos et al., 2014; Xia et
al., 2023; Xue et al., 2022), in line with the similar diurnal patterns of $f$(RH) and SAE
observed in this study. Figure 5 demonstrated a significantly positive correlation
between $f$(RH) and SAE during NPF events, with a similar slope of approximately
0.65 suggesting the consistent variation of $f$(RH) with SAE across both periods. As
larger particles contributed higher to the aerosol volume concentrations (Fig. S3), the
decrease of SAE also corresponded to an increase in $\sigma_{sca,\ 525}$ (Fig. 5a2, b2). In this
sense, $f$(RH) increased with SAE whereas decreased with $\sigma_{sca,\ 525}$, or rather the
pollution level during NPF events. Meanwhile, the cleaner environment of P2 period
generally possessed a lower CS (Table S2, as denoted by the size of circles in Fig. 5),
thereby in favor of the occurrence of NPF event. Such a positive (negative)
correlation of $f$(RH) with SAE (CS) was more pronounced in heatwave-induced high
temperature days during P2 period. The possible reasons can be attributed to the
following two aspects. One is related to the relatively smaller aerosol $R_{eff}$ (with a
larger SAE) due to the lower GR, likely influenced by the evaporation of
newly-formed unstable clusters and particle coatings under heatwaves (Bousiotis et al.,
2021; Cusack et al., 2013; Deng et al., 2020) during the subsequent growth of aerosols.
Secondly, the higher temperature was normally associated with stronger
photochemical oxidation, which could intensify the formation of secondary aerosol
components with a higher hygroscopicity (Asmi et al., 2010; Gu et al., 2023; Liu et al.,
2014; Wu et al., 2016; R. Zhang et al., 2015; Zhang et al., 2024). This is further
supported by the relatively higher levels of UVB (P1: 2.6 ± 1.9 W·m$^{-2}$ versus P2: 2.7
± 2.0 W·m$^{-2}$) and $O_3/O_X$ (P1: 0.81 ± 0.17 versus P2: 0.82 ± 0.17) during P2 heatwave
days, also in line with a recent study which demonstrated that heatwaves affected
secondary organic aerosols (SOA) formation and aging by accelerating
photooxidation in Beijing (Zhang et al., 2024).

It is worth noting that $f$(RH) did not show a consistently higher level after the
NPF occurrence during P2 period, and it was slightly higher within the first few hours



of NPF occurrence during P1 NPF events (Fig. 3b). In fact, aerosol optical
hygroscopicity not fully corresponds to the bulk hygroscopicity primarily determined
by aerosol chemical components, and the variability in aerosol optical features also
plays a key role in $f$(RH). In this sense, the size-dependency of aerosol optical
properties should be considered. The size-resolved $\sigma_{sca,\,525}$ distribution and
size-resolved cumulative frequency distribution (CFD) of $\sigma_{sca,\,525}$ over different NPF
events were calculated using the Mie theory, with good agreements between the
theoretically calculated and measured $\sigma_{sca,\,525}$ values ($R^2 = 0.99$). As shown in Fig. S7
and Fig. S9, new particles must grow into the accumulation mode size at least before
they can exert a significant influence on the total scattering coefficient. The critical
sizes corresponding to the cumulative frequency of 50% in $\sigma_{sca,\,525}$ were 358.7 nm and
333.8 nm on P1 and P2 NPF event days, respectively. This indicates that relatively
smaller particles contributed a slightly higher portion to $\sigma_{sca,\,525}$ during P2 NPF events,
while the $\sigma_{sca,\,525}$ of P1 NPF events was mainly contributed by larger particles.
Nevertheless, the Mie theory suggests that these smaller particles generally have a
weaker enhancement in total scattering after hygroscopic growth, in comparison to
larger size particles (Collaud Coen et al., 2007, Fig. S7). Consequently, the changes in
aerosol optical and hygroscopic properties necessitate consideration of both aerosol
optical and chemical characteristics during different NPF events. The contribution of
newly formed ultrafine particles to aerosol optical properties was insignificant within
the first few hours of NPF occurrence, leading to a reduced enhancement in aerosol
light scattering as characterized by a smaller $R_{eff}$ during P2 NPF events in comparison
to P1 NPF events. In contrast, the growth of pre-existing and newly formed particles
into larger sizes would subsequently affect bulk aerosol optical properties, which was
evidenced by the enhancement in aerosol extinction coefficient observed after NPF
occurrence in a recent study (Sun et al., 2024). Specifically, particles could undergo a
longer and more intensified photochemical aging process during P2 NPF events as
influenced by persistent heatwaves, which facilitated the secondary formation of
hygroscopic aerosols and resulted in a higher $f$(RH) after 15:00 (Fig. 3b).



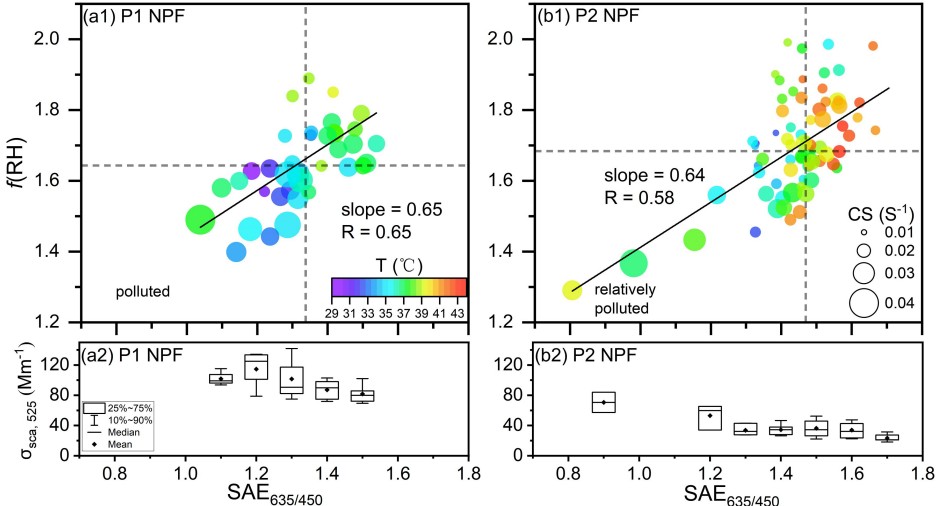

**Figure 5. (a1)** The relationship between $f$(RH) and $SAE_{635/450}$, as well as temperature (as indicated by the color of dots) and CS (as denoted by the size of circles), on P1 NPF event days during the 08:00-22:00 time window. The vertical (horizontal) dash line represents the median value of $SAE_{635/450}$ ($f$(RH)). **(a2)** The corresponding $\sigma_{sca,\,525}$ under different $SAE_{635/450}$ levels on P1 NPF event days. **(b1-b2)** The same but for P2 NPF event days.





### 3.5 $f$(RH)-induced changes in aerosol direct radiative forcing


The changes in $f$(RH) have significant implications for aerosol direct radiative
forcing. Despite considerably lower $\sigma_{sca, 525}$ results during heatwaves, the
corresponding mean $f_{RF}$(RH) levels particularly for P2 NPF event days were higher
than that of the P1 cases (Fig. 6a). A robust positive correlation ($R^2 = 0.68$) was
observed between $f$(RH) and aerosol radiative forcing enhancement factor, $f_{RF}$(RH)
(Fig. 6b). This is likely attributed to the enhanced $f_{RF}$(RH) with the larger forward
scattering ratio β, or rather higher HBF for smaller particle sizes, as supported by a
generally negative correlation between $f_{RF}$(RH) and $R_{eff}$. Specifically, the highest
$f_{RF}$(RH) value of $2.2 \pm 0.2$ was observed on P2 NPF event days, characterized with the
highest $f$(RH) and smallest $R_{eff}$ (i.e., highest HBF) of the entire study period.
The definition of $f_{RF}$(RH) in Eq.(5) implies the dependences of $f_{RF}$(RH) on both
$f$(RH) and HBF-derived β(RH) and β(dry), or rather the ratio of $HBF_{525, RH}/HBF_{525}$.
The mean $HBF_{525, RH}$ was generally larger than $HBF_{525}$ in this study, specifically with
the $HBF_{525, RH}/HBF_{525}$ ratios centered around 1.8 and even approached 2.5 on P2 NPF
event days (Fig. 6c, Table S2). This could be different from the classical Mie theory
with the spherical-particle premise, i.e., the observed light backscattering was
enhanced after hydration likely resulted from the evolution in particle morphology
that significantly influences their optical properties (Mishchenko 2009). The
organic-rich particles might remain non-spherical even after water uptake due to the
efficient evaporation of organic coatings under extremely hot weather conditions, as
evidenced by a recent study that high temperature and RH conditions could accelerate
the evaporation rate of SOA (Li et al., 2019). Meanwhile, the backward scattering
intensity of non-spherical particles is suggested to be much larger than its spherical
counterparts at scattering angles between 90° and 150° (Mishchenko 2009; Yang et al.,
2007). Furthermore, ultrafine particles would significantly contribute to both total
light scattering and backscattering coefficients (Fig. S7) after hygroscopic growth, if
the aerosol population was large enough (e.g., during NPF processes). These
combined effects could potentially change particle morphology and optical properties



(e.g., elevated the $HBF_{525,\,RH}$) particularly during heatwave-influenced NPF events,
characterized with the smallest aerosol $R_{eff}$ (102.8 ± 12.4 nm), lowest number fraction
of accumulation mode particles (0.20 ± 0.10), and a higher SOC/OC ratio. The higher
$HBF_{525,\,RH}/HBF_{525}$ ratios increased the HBF-derived β(RH)/β(dry) levels, in
combination of the elevated $f$(RH), further resulting in the highest $f_{RF}$(RH) observed
during P2 NPF events. Given that previously observed $HBF_{525,\,RH}$ was typically lower
than $HBF_{525}$ (Titos et al., 2021; Xia et al., 2023; L. Zhang et al., 2015), the mean
$f_{RF}$(RH) results of this study ($f_{RF}$(85%) = 2.0 ± 0.2) were significantly higher than
those observed in the Yangtze River Delta ($f_{RF}$(85%) = 1.5, L. Zhang et al., 2015), the
North China Plain ($f_{RF}$(80%) = 1.6 ± 0.2, Xia et al., 2023), and some other regions in
the world (Titos et al., 2021, Fig. 6d). It should be noted that the reported $f_{RF}$(RH) for
the UGR site (Spain) was even higher, likely due to the relatively larger HBF in that
area (Titos et al., 2014; 2021).

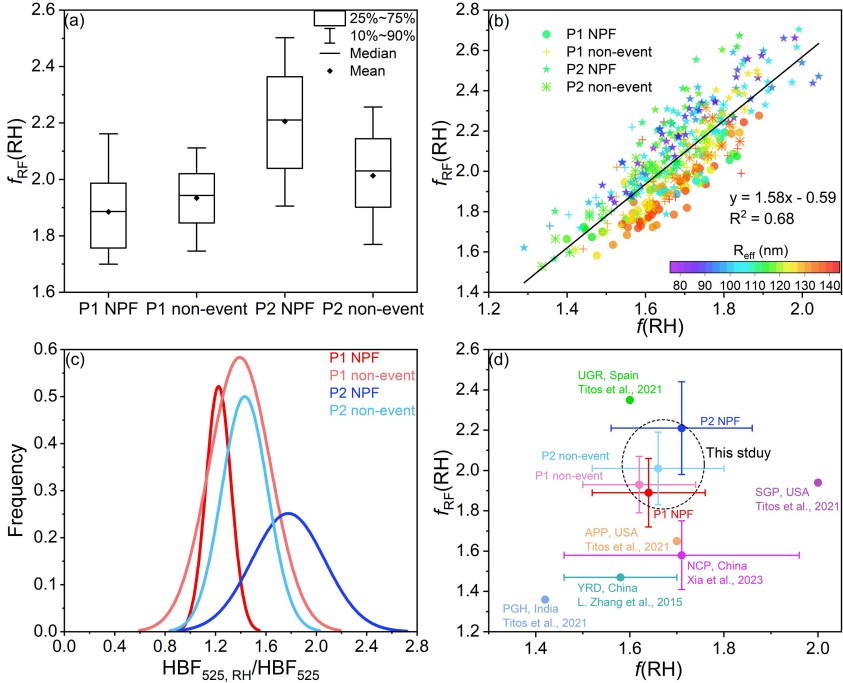


**Figure 6. (a)** The box-plot of $f_{RF}$(RH) during P1 or P2 NPF event and non-event days.
**(b)** The relationship between $f_{RF}$(RH) and $f$(RH), as colored by the corresponding $R_{eff}$,
during P1 or P2 NPF event and non-event days (shown in different symbols). **(c)**





Occurrence frequency of the ratio $HBF_{525,\ RH}/HBF_{525}$ during P1 or P2 NPF event and
non-event days. **(d)** The mean $f_{RF}(RH)$ under different $f(RH)$ levels (the error bars
stand for ± one standard deviations corresponding to $f_{RF}(RH)$ and $f(RH)$, respectively),
along with the reported $f_{RF}(RH)$ and $f(RH)$ data for other regions in the world.
A recent study has indicated that continuous reduction of $PM_{2.5}$ mass loadings
can increase the net solar radiation, thereby promoting NPF events (Zhao et al., 2021).
Given the complexity and dynamic evolution of the atmospheric environment, these
can further alter the intrinsic properties of aerosol particles (e.g., $f(RH)$, HBF,
morphology), potentially feeding back into aerosol-radiation interactions. Our
findings suggest that NPF and growth events may elevate aerosol optical
hygroscopicity in rather hot environments, e.g., the Basin area and tropical regions.
Meanwhile, NPF serves as a crucial secondary transformation process in the
atmosphere (Zhu et al., 2021). The favorable atmospheric diffusion capability ensured
the mixing of newly formed particles into the upper boundary layer, where is colder
and more humid than that near the surface during heatwaves (Jin et al., 2022). Hence,
the enhancement of aerosol optical hygroscopicity during the subsequent growth of
pre-existing and newly formed particles possibly exacerbates secondary pollution and
even triggers haze events (Hao et al., 2024; Kulmala et al., 2021). On the other hand,
the new particles of higher hygroscopicity could contribute more to the activation of
CCN, thereby modulating the aerosol-cloud interactions and further the global climate
(Ren et al., 2021; Sun et al., 2024; Wu et al., 2015). Additionally, the simultaneous
decrease in aerosol effective radius and possibly evaporation-induced non-spherical
particle morphology further enhance the aerosol direct radiative forcing enhancement
factor, potentially amplifying the cooling effect mainly caused by light scattering
aerosols. This highlights the needs for further in-depth exploration on aerosol
radiative impacts at weather extremes (e.g., heatwaves) with the changing climate,
given the continuous reductions of anthropogenic emissions and more intense
emissions of biogenic origins with the global warming. Besides, more detailed
information on the evolution of particle morphology with the changing environment
(e.g., varied temperature and RH) would enrich insights into the aerosol radiative



forcing.

### 4 Conclusions and implications

NPF events frequently occurred in urban Chongqing of southwest China in the
summer of 2022, accompanied with continuous heatwaves. Concurrent measurements
of aerosol optical and hygroscopic properties, PNSD, and bulk chemical compositions
were conducted to elucidate the mechanisms behind the variations in aerosol optical
hygroscopicity during different NPF event and non-event days.
NPF events exhibited distinct characteristics during the normally hot (P1,
relatively polluted) and heatwaves-dominated (P2, quite clean) periods. NPF within
P1 period was favored by the decrease in background aerosol loading and the higher
abundance of $H_2SO_4$. NPF events that occurred during the heatwave P2 period were
characterized with relatively lower CS, CoagS, and GR, as well as a smaller $R_{eff}$, than
P1 NPF cases. In comparison to the P1 NPF events, heatwaves initiated NPF earlier
and prolonged the subsequent growth during P2, likely intensifying the photochemical
oxidation due to heatwave-induced aging processes and modulating the evolution of
aerosol size distributions differently.
Heatwaves also significantly influenced the aerosol optical and hygroscopic
properties. Distinct diurnal patterns of $\sigma_{sca,\ 525}$ were observed for different types of
NPF events, with a minor $\sigma_{sca,\ 525}$ noontime peak occurred in P2 instead of peaked
earlier around the morning rush hours on P1 NPF event days. HBF and SAE were
significantly higher on P2 NPF event days, primarily due to the relatively smaller $R_{eff}$
for heatwave-influenced NPF cases. $f$(RH) remained relatively stable during the
daytime of NPF event days and peaked around 16:00-18:00, likely due to the
intensive photochemical reactions and accordingly enhanced formation of more
hygroscopic secondary aerosols. These secondary components could be more
abundant due to heatwave-induced stronger photooxidation, further resulting in a
higher $f$(RH) particularly during the subsequent growth of pre-existing particles and



newly formed ultrafine ones during P2 NPF events in comparison to that of P1 NPF
cases.

607   Compared with non-event cases, the generally higher levels of daily mean $f$(RH)

suggested that the aerosol optical hygroscopicity was enhanced during NPF events in
hot summer of urban Chongqing. A significantly positive (negative) correlation
between $f$(RH) and SAE (CS, $\sigma_{\text{sca, 525}}$, or rather the pollution level) was observed for
both periods, with a more pronounced correlation during heatwave-influenced NPF
events. The aerosol light scattering or volume concentration was mainly contributed
by the larger accumulation-mode particles, while more ultrafine particles dominated
the size distribution especially for the initial stage of heatwave-influenced NPF events,
further leading to a diminished aerosol scattering enhancement capability in
comparison to P1 NPF events.

617   Changes in $f$(RH) have significant implications for the aerosol direct radiative

forcing. A robust positive (negative) correlation existed between $f_{\text{RF}}$(RH) and $f$(RH)
($R_{\text{eff}}$). Despite a lower $\sigma_{\text{sca, 525}}$ during heatwaves, the corresponding mean $f_{\text{RF}}$(RH) was
relatively higher and the maximum value of $2.2 \pm 0.2$ was observed on P2 NPF event
days, associated with the highest $f$(RH) ($1.7 \pm 0.2$), smallest $R_{\text{eff}}$ ($102.8 \pm 12.4$ nm),
and highest $\text{HBF}_{525, \text{RH}}/\text{HBF}_{525}$ ratios ($1.8 \pm 0.3$). The above highlights that heatwaves
could influence the NPF and atmospheric processing (although with a decreased
aerosol effective radius likely due to evaporation-resulted non-spherical particle
morphology under persistently high temperature conditions), thereby enhancing
aerosol optical hygroscopic growth and potentially reducing the net solar radiation
directly especially in hot summer. Further explorations on detailed molecular
characterizations and aerosol radiative impacts including the aerosol-cloud
interactions of weather extremes (e.g., heatwaves) with the changing climate are
highly recommended.

**Data availability.** Data will be available upon request.



**Author contributions.** YH and PL: Methodology, Investigation, Data analysis,
Formal analysis, Visualization, Validation, Writing – original draft & editing. YG and
ZW: Methodology, Investigation, Formal analysis. MT, YC, HX and WH: Data
curation, Methodology. FW and YL: Investigation. YK: Methodology, Data analysis,
Writing – review & editing. JC: Conceptualization, Methodology, Funding acquisition,
Data curation, Writing – review & editing, Supervision.

**Competing interests.** The authors declare no competing financial interest.

**Acknowledgement.** This work was supported by the National Natural Science
Foundation of China (No. 42105075), Venture and Innovation Support Program
for Chongqing Overseas Returnees (No. cx2021021). We thank Biao Xue for the
technical support on the utilization and maintenance of the humidified
nephelometer system. We also thank Ziqian Wang for the TSP filter sample
collection and Jiawei Zhou for the corresponding offline chemical analysis.

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
