# Peer review of "Divergent changes in aerosol optical hygroscopicity and new particle"

_EGUsphere, 2024_

## Author Response (AR1)

Dear Editor,

We would like to thank all the reviewers for their constructive comments. The insightful suggestions have been fully considered in the revised manuscript. Point-by-point responses to the suggestions, corresponding updates with the revised manuscript, and the finalized version have been uploaded.

In the following, original suggestions, our response, and updates on the revised manuscript are shown in **bold**, normal, and *italic*, respectively.

Kind Regards,

Jing Chen, Yuhang Hao, and Peizhao Li

**Anonymous Referee #2**

**General comments:**

The authors present simultaneous measurements of particle number size distributions, aerosol optical and hygroscopic properties, and bulk chemical composition from urban Chongqing during the extremely hot summer of 2022 to investigate the characteristics of new particle formation (NPF) events for two distinct cases: polluted and clean-heatwave event. The authors claim that heatwave(s) may induce stronger photooxidation, enhancing hygroscopic growth and thereby aerosol direct radiative forcing. Overall, the manuscript is well-written. Although the objective of this study is intriguing, I find that a single heatwave event or unusually hot summer is not necessarily sufficient to support the findings and therefore speculation or tall statements must be avoided. I would like to recommend the publication of this study after the authors carefully address all the following concerns.

**Response:** We thank the reviewer for the insightful comments and supportive recommendation on this manuscript.

**Specific comments:**

RC1. Fig. 1c shows the time evolution of (hourly?) temperature during the study period. Air temperature (RH) steadily increased (decreased) after 8 Aug. Surprisingly, the wind speed is slightly higher during period P2 (heatwave. Fig 2i), but heatwaves are usually associated with stagnant conditions. Nairn et al. (2015) calculated the excess heat factor to identify heatwave events. Could this be explored to determine the spatial extent of this particular heatwave event, using gridded temperature data if it is available for the region? Heatwaves are anomalous events characterized by extremely high surface air temperatures, typically lasting over a week. A mere surface air temperature threshold is not the best indicator of a regional heatwave event. This is indeed critical in the content of regional NPF events and the conclusions drawn from this study. The question is — Did the heatwave event trigger the NPF event, or did relatively cleaner conditions favour the NPF event or specific dynamical weather pattern favoured NPF (high-pressure system) or a combination of everything?

Response: Figure 1c depicts the temporal variation of hourly temperature records during the study period. Although slightly higher than that of P1 (Table S2), the mean level of wind speed during P2 is still within 2.0 m/s, i.e., the gentle breeze condition. The 2022 summer in China was demonstrated to be characterized by an unprecedented heatwave event, as evidenced by regional gridded temperature results from recent studies (Chen et al., 2024; Wang et al., 2024). These studies have illustrated the widespread distribution of elevated temperature levels across multiple regions in China during August 2022 compared to the same periods in previous years, with particularly intense heatwave impacts observed in Southwest China. This robust evidence confirms that our study period was indeed influenced by a severe and extensive heatwave event.

Following the reviewer's suggestion, we have calculated the corresponding Excess Heat Factor (EHF) metric (Nairn and Fawcett, 2014) for the study period (Figure R1). The EHF results suggest that the period following 9 August 2022 belongs to heatwave cases, aligning with the observed meteorological conditions. In addition

to the EHF analysis, China Meteorological Administration normally defines heatwaves as follows: three or more consecutive days with a daily maximum temperature,  $T_{max}$ , exceeding 35 °C (http://www.cmastd.cn/standardView.jspx?id=2103) (Guo et al., 2016; Sun et al., 2014; Tan et al., 2007). According to the above criteria and the consistently occurrence of  $T_{max} \geq 38$  °C (approximately the last 25th percentile of temperature records for the whole observation period, Figure R1a-b), 7–19 August 2022 was classified as a heatwave-dominated period in this study.

Figure R1. (a) Time series of calculated EHF, along with the daily maximum temperature ( $T_{max}$ ) and dry  $\sigma_{sca, 525}$  results, during the study period. The corresponding occurrence frequency and cumulative frequency of hourly (b) temperature and (c)  $\sigma_{sca, 525}$  data records.

On the other hand, we agree that NPF events during different periods are not solely influenced by temperature/solar radiation variations but also regulated by other factors (e.g., pollution levels). The environment was significantly cleaner after August 6, as evidenced by the fact that the hourly mean  $\sigma_{sca, 525}$  values were exclusively below

100 Mm-1 (approximately the last 10th percentile of dry aerosol light scattering data, regarded as the threshold value of relatively polluted cases; Figure R1c). In this sense, we defined the period from July 29 to August 6 as a relatively polluted hot period (P1; NPF during this period was marked as NPFp), whereas a clean and heatwave-dominated period from August 7-19 (P2; NPF was accordingly labelled as NPFC, HW). This dual consideration of temperature and pollution levels could provide a more holistic understanding of the mechanisms of NPF events and related environmental and climatic impacts. To avoid arbitrary statements, we emphasize that heatwave-induced atmospheric conditions diverge significantly from that of normal periods, as characterized by the observed higher (lower) solar radiation (RH), along with the potential changes in types and concentrations of gaseous precursors in this study.

Accordingly, we have revised the manuscript to underscore the distinct changes in NPF events and aerosol optical hygroscopicity against the background of heatwaves, thereby deepening insights into the implications of heatwave conditions for aerosol physicochemical properties.

Updated on the manuscript:

**Abstract:** Compared to the  $NPF_P$  events,  $NPF_{C, HW}$  occurred approximately one hour earlier and the subsequent growth was prolonged, accompanied by a smaller aerosol effective radius ( $R_{eff}$ ) and lower formation/growth rate during heatwaves.

L120-123: During the summer of 2022, a rare heatwave event raged throughout China, especially the Sichuan-Chongqing region of southwest China (Chen et al., 2024; Wang et al., 2024), with the daily maximum temperature exceeding 40 °C lasted for 29 days observed at Beibei meteorological station in Chongqing (Hao et al., 2023).

L142-155: During the observation period, urban Chongqing suffered a rare heatwave (Fig. S1; Chen et al., 2024; Wang et al., 2024), which significantly affected the local transportation and industrial activities (Hao et al., 2023). China Meteorological Administration (CMA) defines heatwaves as three or more consecutive days with daily maximum temperature ( $T_{max}$ ) above 35 °C

(http://www.cmastd.cn/standardView.jspx?id=2103; Guo et al., 2016; Sun et al., 2014; Tan et al., 2007). Since no unified definition of heatwaves worldwide, the whole study period was categorized into two stages according to CMA's criteria of the daily  $T_{max}$  records and the Excess Heat Factor (EHF) metric proposed by Nairn and Fawcett (2014) (Fig. S2a): (1) the normally hot period from 29 July to 6 August (marked as P1); (2) the heatwave-dominated period from August 7-19 (marked as P2) characterized with the consistently occurrence of  $T_{max}$  exceeding 38 °C (approximately the last  $25^{th}$  percentile of temperature records for the whole observation period; Fig. S2b).

L313-315: Additionally, the mean CS of the NPFP events was above 0.015 S-1 (Table S2), which could be considered as the "polluted" NPF day (Shang et al., 2023).

L353-354: The NPF events under heatwaves usually initiated earlier (Fig. S8),

**L648-649:** In comparison to the P1 NPFP events, NPFC, HW occurred approximately one hour earlier and the subsequent growth was longer during P2,

**Updates in the reference list:**

Chen, T., Wang, T., Xue, L., and Brasseur, G.: Heatwave exacerbates air pollution in China through intertwined climate-energy-environment interactions, Sci. Bull., 69, 2765–2775, https://doi.org/10.1016/j.scib.2024.05.018, 2024.

Nairn, J. R. and Fawcett, R. J. B.: The excess heat factor: A metric for heatwave intensity and its use in classifying heatwave severity, Int. J. Environ. Res. Public Health, 12, 227–253, https://doi.org/10.3390/ijerph120100227, 2014.

Sun, X., Sun, Q., Zhou, X., Li, X., Yang, M., Yu, A., and Geng, F.: Heat wave impact on mortality in Pudong New Area, China in 2013, Sci. Total Environ., 493, 789–794, https://doi.org/10.1016/j.scitotenv.2014.06.042, 2014.

Tan, J., Zheng, Y., Song, G., Kalkstein, L. S., Kalkstein, A. J., and Tang, X.: Heat wave impacts on mortality in Shanghai, 1998 and 2003, Int. J. Biometeorol., 51, 193–200, https://doi.org/10.1007/s00484-006-0058-3, 2007.

Wang, N., Du, Y., Chen, D., Meng, H., Chen, X., Zhou, L., Shi, G., Zhan, Y., Feng, M., Li, W., Chen, M., Li, Z., and Yang, F.: Spatial disparities of ozone pollution in the Sichuan Basin spurred by extreme, hot weather, Atmos. Chem. Phys., 24, 3029–3042, https://doi.org/10.5194/acp-24-3029-2024, 2024.

Updated on the Supplement:

The aforementioned Figure 1R has also been added into the Supplement (i.e., Figure S2):

L42-44: Based on the method proposed by Nairn and Fawcett (2014), the Excess Heat Factor (EHF) metric was accordingly calculated for this study (Figure S2a).

RC2. Please provide statistics of NPF events and non-events for both periods. A total of 23 days is divided into 4 categories and conclusions are drawn from a mere one heatwave event. How confidently can you say heatwave(s) promote NPF (based on your results alone)? How about NPF frequency from previous years during the same time period? I also suggest showing an averaged contour plot of particle number size distributions for all these four categories.

**Response:** We thank the reviewer for this valuable suggestion. As stated in our response to **RC1**, our study is attempted to stress that atmospheric conditions were significantly impacted under the heatwave weather, further influencing the corresponding NPF and aerosol physicochemical properties. We have followed the suggestion and provided the frequency results of NPF days, non-event days and undefined days during both periods in Figure S4a, and the averaged contour plots of PNSD, as well as Dmode, Reff, and CS for these four categories were given in Figure S6.

Due to the data availability, we have compared the PNSDs measured at the same site in summer of 2023, which was similarly divided into P12023 and P22023 periods with the same dates as that of summer 2022. The frequencies of NPF events during P12023 (mean T: 30.5  $\pm$  3.3 °C) and P22023 (mean T: 32.9  $\pm$  3.1 °C) in 2023 were generally identical to that for 2022 (Figure R2), likely suggesting that heatwaves did

not significantly change the occurrence of NPF events. However, the mean star time and growth end time of NPFc, HW events were 9:36 LT (11:07 LT) and 20:37 LT (19:20 LT), respectively, during the P2 period (P22023 period) (Figure R4). This signifies that NPFC, HW events occurred earlier and the subsequent growth was prolonged during the P2 heatwave-dominated period. Besides, the impacts of heatwaves on the subsequent growth of NPF events were rather evident. For instance, aerosol Reff was much smaller on the P2 NPFC, HW days. As shown in Figure S6, the Reff and particle Dmode nearly kept at a same level below/approaching 50 nm during the subsequent growth on the P2 NPFC, HW days. Differently, the Reff was generally above 50 nm and larger than Dmode for both P1 NPFP cases and non-event days. Moreover,  $R_{\rm eff}$  for  $P2^{2023}$  NPF days (123.9  $\pm$  9.5 nm) were significantly higher than those for P2 NPFC, HW days (102.8  $\pm$  12.4 nm) (Figure R3b2), whereas comparable to that of the P1 NPFP days (124.8  $\pm$  10.7 nm). These differences highlight the uniqueness of P2 NPFC, HW events affected by heatwaves despite of insignificant discrepancy in the occurrence frequency of NPF, which merits more in-depth exploration. Further studies on the mechanisms underlying these changes, particularly under extreme weather conditions, are therefore highly recommended.

**Figure R2.** The occurrence frequencies of NPF, non-event and Undefined days during P1 ( $P1^{2023}$ ) and P2 ( $P2^{2023}$ ) periods of summer 2022 (2023).

Figure R3. The diurnal variations of T and RH (a1), as well as  $R_{eff}$  (b1) on the P2 NPFC, HW and  $P2^{2023}$  NPF days, and the corresponding box plots of T, RH (a2) and  $R_{eff}$  (b2).

**Figure R4.** The start and end time of NPF, along with the subsequent growth end time and their corresponding temperature levels during NPF events in summer 2022 and the same period in summer 2023.

Considering the above aspects, we have updated the manuscript as follows:

**L271-273:** The particle number size distribution data suggested that NPF events appeared in about half the number of observation days (Fig. 1i), with an overall occurrence frequency of 52.4% (Fig. S4a).

**L349-350:** To further investigate the effect of heatwave on NPF events, the diurnal variations of PNSD,  $R_{eff}$  and particle mode diameter ( $D_{mode}$ ) are shown in Fig. S6.

**L353-357:** The NPF events under heatwaves usually initiated earlier (Fig. S8), with the number concentration of nucleation mode particles ( $N_{Nuc}$ ) in P2 NPFC, HW cases peaked about an hour earlier in comparison to NPFP days (Fig. S7a). The  $D_{mode}$  on P2 NPFC, HW days also reached its minimum earlier than that on P1 NPFP days (Fig. S6).

L369-379: In addition, aerosol  $R_{eff}$  was significantly smaller on the NPFC, HW days under heatwave conditions. The  $R_{eff}$  and  $D_{mode}$  nearly kept at a same level below/approaching 50 nm during the subsequent growth on the P2 NPFC, HW days, while the  $R_{eff}$  was generally above 50 nm and larger than  $D_{mode}$  for both P1 NPFP cases and non-event days (Fig. S6). The diurnal patterns of aerosol volume concentrations for different size modes were similar to that of aerosol number concentrations during NPF events (Fig. S7b1-b3). However, both the  $R_{eff}$  of Aitken mode particles ( $R_{Ait}$ ) and accumulation mode particles ( $R_{Acc}$ ) were smaller during P2 NPFC, HW events than that of P1 NPFP events (Fig. S7c2-c3), which may further influence size-dependent aerosol optical and hygroscopic properties (e.g.,  $\sigma_{sca, 525}$ , HBF, SAE, f(RH)).

Updated on the Supplement:

L131-133: The specific dates for NPF and non-event classifications were summarized in Table S1, and the frequencies of NPF, non-event and Undefined days during both periods were shown in Figure S4a.

Figure S4. (a) The occurrence frequencies of NPF, non-event and Undefined days during P1, P2 and the whole observation periods. (b) The 48-h air-mass back trajectories during the study period.

**Figure S6.** Diurnal variations of PNSDs,  $D_{mode}$ ,  $R_{eff}$ , and CS during P1 and P2 NPF days (a1, b1) and non-event days (a2, b2), the error bars stand for  $\pm$  one standard deviations.

RC3. Air mass history plays also a critical role in new particle formation processes. Consider showing an airmass history analysis (source and altitude) using HYSPLIT or Flexpart or similar models. The wind direction during the P2 period appears to be persistently east-southeast.

Response: We agree that air mass history could facilitate the analysis on NPF events. As suggested, the 48-h back trajectories of air mass at 500 m altitude above the site during this study period were calculated using the HYSPLIT (Hybrid Single-Particle Lagrangian Integrated Trajectory) 4 model developed by NOAA (Stein et al., 2015). Results were visualized by MeteoInfoMap (version 3.9.9) (Chen et al., 2021; Tian et al., 2021; Wang, 2014), as shown in the above Figure S4b. The predominant southerly breeze during the summer campaign likely suggests that some other factors could affect the NPF events more significantly in P1 and P2 periods.

We have added the discussion on back trajectories analysis in the revised manuscript.

**In the main text:**

**L205-207:** The descriptions of simultaneous meteorological and air quality data can be found in Sect. S4, and the 48-h backward trajectory analysis was given in Sect. S5 of the supplement.

L297-302: The backward trajectory analysis revealed that the southerly breeze was predominant during the study period (Fig. S4b). Although the surface wind vector slightly varied between the P1 and P2 periods, this consistency in air mass origins suggests that some other factors (e.g., changes in environmental conditions and emissions of gaseous precursors under heatwaves) could have played a crucial role in modulating NPF events.

**In the Supplement:**

L133-137: By using the HYSPLIT (Hybrid Single-Particle Lagrangian Integrated Trajectory) 4 model developed by NOAA (Stein et al., 2015), the 48-h back trajectories of air masses at 500 m altitude above the observation site during this study period were calculated and visualized by MeteoInfoMap (version 3.9.9; Figure S4b) (Chen et al., 2021; Tian et al., 2021; Wang, 2014).

**Updates in the reference list of Supplement:**

Chen, J., Wu, Z., Chen, J., Reicher, N., Fang, X., Rudich, Y., and Hu, M.: Size-resolved atmospheric ice-nucleating particles during East Asian dust events,

Atmos. Chem. Phys., 21, 3491–3506, https://doi.org/10.5194/acp-21-3491-2021, 2021.

Stein, A. F., Draxler, R. R., Rolph, G. D., Stunder, B. J. B., Cohen, M. D., and Ngan, F.: Noaa's hysplit atmospheric transport and dispersion modeling system, Bull. Am. Meteorol. Soc., 96, 2059–2077, https://doi.org/10.1175/BAMS-D-14-00110.1, 2015.

Tian, J., Guan, H., Zhou, Y., Zheng, N., Xiao, H., Zhao, J., Zhang, Z., and Xiao, H.: Isotopic source analysis of nitrogen-containing aerosol: A study of PM2.5 in Guiyang (SW, China), Sci. Total Environ., 760, 143935, https://doi.org/10.1016/j.scitotenv.2020.143935, 2021.

Wang, Y. Q.: MeteoInfo: GIS software for meteorological data visualization and analysis, Meteorol. Appl., 21, 360–368, https://doi.org/10.1002/met.1345, 2014.

RC4. How is aerosol optical enhancement factor related to particle diameter for both cases (RH

**Figure 4.** Diurnal variations of **(a)** the number fraction (NFAcc.) and **(b)** volume fraction of accumulation mode particles (VFAcc.) on P1 (red) and P2 (blue) NPF days (solid line), as well as non-event days (dash line). The time window of 08:00-22:00 LT was shaded in red. The relationship of f(RH) with  $R_{eff}$  and  $VF_{Acc.}$  (as indicated by the colored dots) on P1 **(c)** and P2 non-event days **(d)**, as well as on P1 **(e)** and P2 **(f)** NPF days during the 08:00-22:00 LT time window.

**Figure 5.** The relationship between f(RH) and  $SAE_{635/450}$ , as well as temperature (as indicated by the color of dots, missing values are represented in gray) and CS (as denoted by the size of circles), on P1 non-event days (a1), NPFP days (a2) during the 08:00-22:00 LT time window. The vertical (horizontal) dash line represents the median value of  $SAE_{635/450}$  (f(RH)). (a3) The corresponding  $\sigma_{sca}$ , 525 under different  $SAE_{635/450}$  levels on P1 NPFP days. (b1-b3) The same but for P2 period.

**RC6. Why GR (<25nm, 25-100 and >100 nm) and FR are not reported and compared between the event types based on SMPS data.?**

**Response:** We have performed additional calculations of FR and GR of different size ranges following the methodologies introduced by Kulmala et al. (2012), and the specific results were summarized in Table R1. The results indicate that both FR and GR (<25 nm) of new particles on the P1 NPFP days were generally higher than those for P2 NPFC, HW days. Notably, when aerosol particle sizes are in the range of tens of

nanometers, their sources are significantly influenced by primary emissions and transport processes, which may introduce errors into the calculation results (Shang et al., 2023). Furthermore, when aerosol particle sizes exceed 80 nm, the "maximum concentration method" cannot effectively calculate aerosol growth rates (Dal Maso et al., 2005; Kulmala et al., 2012).

**Table R1.** The FR (Formation Rate) and the GR of different size ranges on the NPF days in this study

| Date | FR (cm -3 s -1 ) | GR<25 nm (nm·h -1 ) | GR 25-40 nm (nm·h -1 ) | GR 40-60 nm (nm·h -1 ) | GR 60-80 nm (nm·h -1 ) |
|------|----------------------------------------|--------------------------------|----------------------------------------------|----------------------------------------------|----------------------------------------------|
| 7.29 | 15.78                                  | 9.98                           | /                                            | /                                            | /                                            |
| 8.1  | 10.06                                  | 14.42                          | /                                            | 1.55                                         | /                                            |
| 8.2  | /                                      | /                              | 5.67                                         | 5.27                                         | 4.81                                         |
| 8.3  | 25.47                                  | 16.63                          | 8.57                                         | 13.78                                        | 16.65                                        |
| 8.7  | 6.22                                   | 10.08                          | 8.79                                         | /                                            | /                                            |
| 8.8  | /                                      | /                              | 15.53                                        | 6.38                                         | 6.22                                         |
| 8.9  | 8.47                                   | 12.09                          | 9.82                                         | 6.21                                         | 1.59                                         |
| 8.12 | 18.98                                  | 5.76                           | 5.86                                         | 3.91                                         | 6.6                                          |
| 8.13 | /                                      | /                              | 8.27                                         | 4.01                                         | 8.3                                          |
| 8.14 | /                                      | /                              | 13.37                                        | 4.21                                         | 2.07                                         |
| 8.19 | /                                      | /                              | 2.88                                         | 1.72                                         | 8.3                                          |

We have added the above mean FR and GR results into the original Table S2, and updated the corresponding discussion in the revised manuscript:

**In the main text:**

**Abstract:** accompanied by a smaller aerosol effective radius ( $R_{eff}$ ) and lower formation/growth rate during heatwaves.

L190-193: representative parameters for NPF events, e.g., the formation rate (FR) and growth rate (GR) of new particle, condensation sink (CS) and coagulation sink (CoagS) (Dal Maso et al., 2005; Kulmala et al., 2012). More details are provided in the supplement (Sect. S5).

**L364-365:** Given that the growth rates of new particles were generally lower during  $P2 NPF_{C, HW}$  events (Table S2),

L382-383: (2) lower FR and GR of particles under the cleaner environment (Table S2);

**L500-502:** One is related to the smaller aerosol  $R_{eff}$  (with a larger SAE) due to the lower FR and GR, likely influenced by the evaporation of newly-formed unstable clusters and particle coatings under heatwaves

**L646-648:** NPFC, HW events that occurred during the heatwave P2 period were characterized with lower CS, CoagS, FR and GR, as well as smaller  $R_{eff}$  and  $D_{mode}$ , than P1 NPFP cases.

**In the Supplement:**

L126-130: Using the measured PNSD data, NPF events were identified according to the criteria raised by Dal Maso et al. (2005), and the key parameters related to NPF events (e.g., formation rate (FR) and growth rate (GR) of new particles, condensation sink (CS) and coagulation sink (CoagS)) could be derived following the methodologies introduced by Dal Maso et al. (2005) and Kulmala et al. (2012).

**Updates in the reference list of Supplement:**

Dal Maso, M., Kulmala, M., Riipinen, I., Wagner, R., Hussein, T., Aalto, P. P., and Lehtinen, K. E. J.: Formation and growth of fresh atmospheric aerosols: Eight years of aerosol size distribution data from SMEAR II, Hyytiälä, Finland, Boreal Environ. Res., 10, 323–336, 2005.

**Technical comments:**

RC7. Abstract: "Heatwaves triggered NPF earlier" – please quantify. You may want to plot sunrise and sunset times in Fig. S5. Define NPF event end time and growth event end time somewhere in the text.

**Response:** As mentioned in our response to **RC1** (Referee #2), we have revised the corresponding statement in the abstract. Given the insignificant variability in both sunrise and sunset times during the study period (i.e., just within half an hour

discrepancy; Figure R5), we have included such information into the main text instead of adding these data in Figure S5. The definitions of the NPF event end time and growth end time were accordingly updated in the Supplementary.

Figure R5. Variations in sunrise and sunset times during the study period.

Updates in the main text:

**Abstract:** Compared to the NPFP events, NPFC, HW occurred approximately one hour earlier and the subsequent growth was prolonged, accompanied by a smaller aerosol effective radius ( $R_{eff}$ ) and lower formation/growth rate during heatwaves.

L357-361: Since the sunrise and sunset time did not significantly vary within the study period (i.e., less than a half hour discrepancy), heatwaves likely provided more favorable conditions (e.g., enhanced volatile gaseous emissions, low RH; Bousiotis et al., 2021; Hamed et al., 2007; Wang et al., 2024) for the occurrence of NPF events in urban Chongqing.

Updates in the Supplement:

L160-169: The specific start and end time of NPF, along with the subsequent growth end time, during NPF events were displayed in Figure S8. The NPF event end time is defined as the moment when the formation of new nucleation-mode particles (diameter <25 nm) ceases, specifically identified by the absence of a notable increase in sub-25 nm particles (Dal Maso et al., 2005; Hamed et al., 2007; Kerminen et al., 2018). The growth event end time refers to the time when the newly formed particles

stop growing, typically due to the depletion of low-volatility vapors or particle coagulation (Dal Maso et al., 2005; Kerminen et al., 2018). This can be observed as the stabilization of particle diameters in the Aitken/accumulation mode, marked by a flattening of the growth trajectory in the PNSD plot (Figure 1i).

**Updates in the reference list of the Supplement:**

Hamed, A., Joutsensaari, J., Mikkonen, S., Sogacheva, L., Dal Maso, M., Kulmala, M., Cavalli, F., Fuzzi, S., Facchini, M. C., Decesari, S., Mircea, M., Lehtinen, K. E. J., and Laaksonen, A.: Nucleation and growth of new particles in Po Valley, Italy, Atmos. Chem. Phys., 7, 355–376, https://doi.org/10.5194/acp-7-355-2007, 2007.

Kerminen, V. M., Chen, X., Vakkari, V., Petäjä, T., Kulmala, M., and Bianchi, F.: Atmospheric new particle formation and growth: Review of field observations, Environ. Res. Lett., 13, https://doi.org/10.1088/1748-9326/aadf3c, 2018.

RC8. Consider an obvious abbreviation for event classification – relatively polluted period (P1) to be indicated as  $NPF_{polluted}$  and clean heatwave-induced to be indicated as  $NPF_{clean,\,HW}$

**Response:** We have accepted this suggestion and updated throughout the manuscript and figures (i.e., NPFP and NPFC, HW):

**L284-287:** Correspondingly, NPF events occurring during the relatively polluted P1 period (as detailed in section 3.2) were defined as NPFP, while cases during the cleaner and heatwave-dominated P2 period were classified as NPFC, HW.

RC9. All figure captions should clearly mention what is being plotted, time resolution, time (local to UTC), etc and they should be self-explanatory.

**Response:** We have updated all the figures, and all the time in the text was labelled as local time (LT).

RC10. There is an interesting recent paper by Garmash et al., 2024, the authors should consider citing and discussing it – DOI 10.1088/1748-9326/ad10d5

**Response:** We appreciate the recommendation of this article, which indeed provides support for our study. We have cited this paper and discussed accordingly in the revised manuscript:

L336-341: However, it is reported that excessive heat can increase the evaporation rate of critical acid-base clusters during the nucleation process and reduce the stability of initial molecular clusters (Bousiotis et al., 2021; Kurtén et al., 2007; Zhang et al., 2012), in line with a recent study that NPF events were weaker during heatwaves in Siberian boreal forest due to the unstable clusters (Garmash et al., 2024).

**L379-382:** The decrease in  $R_{Ait.}$  and  $R_{Acc.}$  during heatwaves could be attributed to three factors: (1) evaporation of the outer layer of particles and unstable clusters due to heatwaves (Bousiotis et al., 2021; Cusack et al., 2013; Deng et al., 2020; Garmash et al., 2024; Li et al., 2019);

**Updates in the reference list:**

Garmash, O., Ezhova, E., Arshinov, M., Belan, B., Lampilahti, A., Davydov, D., Räty, M., Aliaga, D., Baalbaki, R., Chan, T., Bianchi, F., Kerminen, V. M., Petäjä, T., and Kulmala, M.: Heatwave reveals potential for enhanced aerosol formation in Siberian boreal forest, Environ. Res. Lett., 19, https://doi.org/10.1088/1748-9326/ad10d5, 2024.

**RC11. Particle size distribution measurement size range (and number of bins), and time resolution may be mentioned.**

**Response:** We have updated it accordingly in the Supporting Information, S5:

**S5. Particle number size distribution measurements**

During the field observation, every 3-min PNSD and particle volume size distribution (PVSD) was measured by a SMPS, which consisted of a soft X-Ray neutralizer (model 3088, TSI Inc.), a differential mobility analyzer (model 3081, TSI Inc.), and a condensation particle counter (model 3775, TSI Inc.) (Dominick et al., 2018; Rissler et al., 2006). The SMPS was operated at a sheath/sample flow rate of

3.0/0.3 LPM, and the detected size range was 14.1-710.5 nm with 110 size bins. Data inversion of measured particle size distributions was achieved with the Aerosol Instrument Manager software (AIM, TSI Inc.), including the multiple charge and diffusion corrections (Denjean et al., 2015; Rosati et al., 2022).

**Updates in the reference list of Supplement:**

Denjean, C., Formenti, P., Picquet-Varrault, B., Camredon, M., Pangui, E., Zapf, P., Katrib, Y., Giorio, C., Tapparo, A., Temime-Roussel, B., Monod, A., Aumont, B., and Doussin, J. F.: Aging of secondary organic aerosol generated from the ozonolysis of α-pinene: Effects of ozone, light and temperature, Atmos. Chem. Phys., 15, 883–897, https://doi.org/10.5194/acp-15-883-2015, 2015.

Rosati, B., Isokääntä, S., Christiansen, S., Jensen, M. M., Moosakutty, S. P., De Jonge, R. W., Massling, A., Glasius, M., Elm, J., Virtanen, A., and Bilde, M.: Hygroscopicity and CCN potential of DMS-derived aerosol particles, Atmos. Chem. Phys., 22, 13449–13466, https://doi.org/10.5194/acp-22-13449-2022, 2022.

**RC12. How was MLH obtained? All data and methods must be explicitly stated.**

**Response:** We have adjusted the descriptions:

**S4. Meteorological and air quality data**

All the contemporary hourly meteorological datasets including relative humidity (RH), temperature (T), visibility (VIS), wind speed (WS), wind direction (WD), precipitation were obtained from the Integrated Surface Database from the U.S. National Centers for Environmental Information (https://ncdc.noaa.gov/isd) (Wan et al., 2023; Xu et al., 2020), and the mixing layer height (MLH) data were achieved from China Meteorological Administration in this study.

**RC13. Page 4, line 127 -130: consider revising. The data/event sample is too small to draw implications for climate.**

**Response:** We appreciate this suggestion and have updated the Introduction as below:

L134-136: This study will further enrich insights into the potential environmental impacts due to variations in the aerosol optical hygroscopicity and size distribution, specifically under weather extremes (e.g., heatwaves) with the changing climate.

RC14. Page 5, Line 147-149: If I understand correctly, the authors deployed two nephelometers, one with a humidification unit and the other without. I would suggest giving explicit details of how the measurements were conducted.

**Response:** Yes, we deployed two nephelometers in parallel to measure the aerosol scattering coefficients in both dry and wet conditions. We have updated the details to clarify this point:

L161-168: Ambient air was firstly dried through a Nafion dryer (model MD-700, Perma Pure LLC) to ensure RH <35%, then split into two streams for both dry and humidified nephelometers operated in parallel. The flowrate for each nephelometer was 2.6 LPM. The aerosol scattering ( $\sigma_{sca, \lambda}$ ) and backscattering coefficients ( $\sigma_{bsca, \lambda}$ ) were detected in a dry state (RH <35%) and at a controlled RH level of  $85 \pm 1\%$ , respectively, with the humidification efficiency regulated automatically by a temperature-controlled water bath. More details on the home-built humidified nephelometer system are available in Kuang et al. (2017, 2020) and Xue et al. (2022).

RC15. Page 7, Line 212: chemical analysis results are plotted in Fig. S2, correct it.

Response: Updated.

**L194-195:** Results of the offline chemical analysis with TSP filter samples are provided in Sect. S3 and Fig. S3.

RC16. Page 7, Line 220: "Fig." S4? I cannot find meteorological and air quality data. Or do you mean Fig.2? Please check all figures numbering and citations in the text.

**Response:** We have checked and corrected all the figures, texts and tables numbering and citations.

RC17. Page 18, Lines: 443-445, consider revising the sentence starting "In this sense...." What pollution level are you referring to?

**Response:** We have revised it as below:

**L493-496:** Given that larger  $\sigma_{sca, 525}$  values typically indicate the condition of a higher aerosol loading, f(RH) increased with SAE whereas decreased with  $\sigma_{sca, 525}$ , or rather the pollution level, during NPF days.

**RC18. Page 18, Lines:445, Remove "Meanwhile"**

Avoid unnecessary use of "pretty" (page 14, line 351), "relatively", "meanwhile", "In this sense" as above, etc. throughout the manuscript. Also Page 3, line 97 "there have been a great many studies" – looks unnecessary and no study is cited either. Simply say "Previous studies showed...." and cite relevant studies.

**Response:** We have accordingly adjusted the corresponding expressions throughout the manuscript.

RC19. How was the aerosol effective radius calculated? Figures S4c1, c2, and c3 are unclear to me. Further, authors should show how the particle mode diameter behaved during P1 and P2 (averaged diurnal variation) for both event types and the condensation sink

**Response:** The calculation of Reff has been given in S5 in the Supplementary, and the detailed description of aerosol Reff can be found in Hansen and Travi (1974), and Grainger et al. (1995). In brief, Reff is the effective mean radius of the aerosol population that can reflect the influence of aerosol size distribution on the light scattering, which depends on the cross-section of particles per unit volume (Hansen and Travi, 1974). Hence, Reff of the nucleation mode (RNuc.), Aitken mode (RAit.), and accumulation mode (RAcc.) particles in Figures S7c1-c3 can be accordingly calculated with the aerosol volume and surface area concentrations of different mode particles.

We have added the diurnal variations of  $D_{mode}$ ,  $R_{eff}$  and CS for both event and non-event days in Figure S6, as included in the previous response to **RC2** (Referee #2).

RC20. There are several linguistic errors or issues with sentence phrasing. As I am not a native English speaker, I prefer not to correct them for the authors. I kindly urge authors to thoroughly proofread the manuscript to ensure clarity before submission. This will greatly enhance the readability and overall impact of the work.

**Response:** Thanks for the suggestion and we have improved the expression of the manuscript.

**Anonymous Referee #3**

**General comments:**

This manuscript focus on the aerosol optical hygroscopicity in Chongqing

during three weeks' field campaign using a combination of a home-built

humidified nephelometer system and a scanning mobility particle sizer (SMPS),

the total suspended particle (TSP) filter sampling and following chemical

analysis, as well as the air pollutants data and meteorological data from available

sources.

The measured aerosol scattering coefficients, aerosol optical hygroscopicity

f(RH), and particle number size distribution are reported. Based on the

temperature and aerosol scattering data, the measurement period from July 19

to August 19, 2022 was divided into P1 period and P2 period

(heatwaves-dominated). The authors discussed the characteristics of NPF events

during P1 and P2 periods, the characteristics of the aerosol optical and

hygroscopic properties during P1 NPF and P2 NPF events, and the effects of

f(RH) on aerosol direct radiative forcing.

**Response:** Thanks for the comments.

**Major comments:**

RC1. Section 3.3 Characteristics of the aerosol optical and hygroscopic properties during NPF events

As can be seen from Figure S7, Size-dependent light scattering, backscattering and HBF efficiencies showed the particles with diameter less than 100 nm have insignificant contribution to aerosol scattering. If so, why the authors pay more attention on the aerosol optical and hygroscopic properties during NPF events? What the exact meaning of NPF events here? Since there are 4 and 7 NPF days during P1 and P2 period, it is a little bit hard to investigate the influence of heatwaves on NPF events and subsequent impacts on aerosol optical and hygroscopic properties.

**Response:** As clarified in our responses to Referee #2, the primary focus of this study is the significant changes observed in both NPF events and aerosol optical and hygroscopic properties against the background of heatwaves. NPF events during the heatwave-dominated P2 period exhibited distinct differences in comparison to normal summer conditions during P1, as well as to the same period in 2023 (refer to our responses to **RC1 and RC2** of Referee #2, Figures R3, R4). These differences are expected to further influence aerosol optical and hygroscopic properties.

We acknowledge that newly formed ultrafine particles have a weak contribution to the aerosol light scattering, yet the subsequent growth into larger sizes combined with atmospheric aging of both pre-existing and newly formed particles could significantly impact aerosol optical hygroscopicity, f(RH). In this study, we consistently observed that the aerosol f(RH) was higher on NPF days compared to non-event days in both periods. Furthermore, heatwaves can intensify photochemical aging processes and prolonged the subsequent growth of new particles on P2 NPFC, HW days, resulting in an even higher f(RH) than that for NPFP days during P1. This agrees with previous studies that atmospheric conditions which are favorable for the occurrence of NPF can also promote the growth of newly formed particles, thereby enhancing aerosol hygroscopicity (Cheung et al., 2020; Wu et al., 2015, 2116). Additionally, NPF events can increase aerosol extinction coefficients compared to

non-event days and even trigger haze pollution (Kulmala et al., 2021; Shen et al., 2011; Sun et al., 2024; Tang et al., 2021), likely suggesting the potential role of amplified aerosol light extinction ability during NPF events. While generally limited contributions, impacts of the ultrafine-mode particles on aerosol optical hygroscopic properties could become more evident through subsequent particle growth in combination with aging processes of both pre-existing and newly formed particles on NPF days, specifically under heatwave weather. This underlines our motivation for investigating aerosol optical and hygroscopic properties in the context of NPF events under normal and heatwave conditions, which are crucial for understanding the impacts of extreme weather events (e.g., heatwaves) on aerosol physicochemical properties with the changing climate.

We have adjusted the title of Section 3.3 into "Characteristics of the aerosol optical and hygroscopic properties on different types of NPF days" and revised the corresponding expression throughout the manuscript, to avoid the misunderstanding that NPF events affect the aerosol optical hygroscopicity predominantly.

RC2. Section 3.5 f(RH)-induced changes in aerosol direct radiative forcing The effect of aerosol hygroscopicity on aerosol direct radiative forcing depends on f(RH) and the ratio of HBF525, RH to HBF525 ratio which were measured in this study. The authors stated the mean HBF525, RH was generally larger than HBF525 with the ratios centered around 1.8 and even approached 2.5 on P2 NPF event days (Fig. 6c, Table S2). This result is in contrast with previous results such as from Titos et al. 2021, Xia et al 2023 and so on. The study by Titos et al., 2021 showed the  $f_b(RH=85\%)$  were lower than f(RH=85%) based on the data from 22 different sites covering a wide range of site types (Arctic, marine, rural, mountain, urban, and desert). The study of Xia et al 2023 showed the backscatter hygroscopic growth factor was lower than scattering hygroscopic growth factor (Figure 3) based on 2-year measurement in Beijing. This ratio is critical for this conclusion of this section. The authors need to explain why the ratio is so different? Please give more information about the humidified nephelometer operation information such as the time series of the temperature and relative humidity variation inside the nephelometer, the background variation etc.

**Response:** We appreciate the reviewer's critical comments regarding the discrepancy observed in the ratio of HBF525, RH/HBF525 with previous studies. As shown in Figure R6, RH inside the dry (marked as "D") and wet ("W") nephelometers generally maintained stable throughout the study period, with a synchronized fluctuation in the corresponding temperature records. This confirms the stability of the humidified nephelometer system and the reliability of our measurements, thus instrumental artifacts as a cause of the observed "abnormal" HBF ratio could be excluded.

Figure R6. Time series of the (a) temperature, (b) RH inside the "dry" and "wet" nephelometers, and the (c) measured HBF525,  $_D$  and HBF525,  $_W$  during the study period. The NPF days and non-event days were shaded in red and blue, respectively.

Our data revealed that the  $f_b(RH)$  (aerosol backscattering enhancement factor) exceeded the corresponding f(RH) (Figure R7), suggesting that the increase in  $\sigma_{bsca, 525}$  is more pronounced than that of  $\sigma_{sca, 525}$  upon hydration. This phenomenon was particularly noticeable on P2 NPFC, HW days, when HBF525, RH significantly higher than HBF525 (Figure R6c). Additionally, we derived the asymmetry parameter g (gRH), which positively correlates with the aerosol forward scattering (Andrews et al., 2006; Marshall et al., 1995), from the measured HBF525 (HBF525, RH) with the Mie model (Andrews et al., 2006). The gRH were generally smaller than g for the four categories (Figure R7c), implying that the forward (backward) light scattering decreased (increased) after water uptake. This is especially evident on P2 NPFC, HW days, with a much lower level of gRH was observed.

**Figure R7.** (a) The frequency of f(RH) and  $f_b(RH)$  on the NPF days and non-event days during different periods, (b) The relationship between f(RH) and  $f_b(RH)$  in this study, (c) The box plots of the HBF525 (HBF525, RH) derived  $g(g_{RH})$ .

There are two potential reasons for these "unique" phenomena. Firstly, the abundant nucleation mode particles could not significantly contribute to aerosol  $\sigma_{sca, 525}$  and  $\sigma_{bsca, 525}$  during NPF events, even falling below the detection limit of the nephelometer (i.e., 0.3 Mm-1). However, the contributions of these particles to  $\sigma_{sca, 525}$ and especially to  $\sigma_{bsca, 525}$  were amplified upon humidification in the "wet" nephelometer. As shown in Figure S10a (the original Figure S7), even if these hydrated particles remain small (e.g., below 100 nm), their HBF was significantly higher than that of larger ones and consequently elevated the HBF525, RH levels. This is in line with the simultaneous evolution of aerosol size distributions, which suggest that both Reff and Dmode of nucleation-mode particles were almost below/approaching 50 nm on the P2 NPFC,HW days (Figure S6; refer to the response to **RC2** of Referee#2). Secondly, we hypothesize that particle morphology plays a key role in this study. Previous studies have found that backward scattering intensity of non-spherical particles is suggested to be larger (Mishchenko 2009; Yang et al., 2007). Refer to the response to RC10 (Referee #2), particles may be partly evaporated under heatwaves. Additionally, the organic-rich particles might remain non-spherical due to the

efficient evaporation of organic coatings under high temperature conditions (Li et al., 2019), further enhancing the  $\sigma_{bsca, 525}$  after hygroscopic growth.

Given the aforementioned possible reasons and our responses to RC1 and RC2 of Referee #2, the unique phenomena of higher  $f_b(RH)$  observed during the P2 NPFC, HW days are more pronounced. We acknowledge that these mechanisms merit further validation through molecular-level studies. Future research to investigate the changes in particle morphology, aerosol optical and hygroscopic properties under similar extremely high-temperature conditions (e.g., T >38 °C) is therefore highly recommended.

We have updated the manuscript as follows, and the Figure R7c has been added in the Figure S10 (the original Figure S7):

L574-588: Given that the backward scattering intensity of non-spherical particles is suggested to be much larger than its spherical counterparts at scattering angles between 90° and 150° (Mishchenko 2009; Yang et al., 2007) and that the HBF-derived asymmetry parameter (g) normally correlates positively with the aerosol forward scattering (Andrews et al., 2006; Marshall et al., 1995), the generally smaller  $g_{RH}$  results (in comparison to g) confirmed the decrease (increase) in the forward (backward) light scattering after water uptake (Fig. S10b), likely implying the change in the morphological structure of particles. This is particularly evident for P2 NPFC, HW days, with a much lower level of  $g_{RH}$  was observed (Fig. S10b). Another possible reason is that although the abundant newly formed particles were generally optically-insensitive, their contributions to  $\sigma_{SCa}$ , 525 and especially to  $\sigma_{bSCa}$ , 525 could be amplified upon humidification. Namely, even if these hydrated particles remained small (e.g., below 100 nm), their HBF was significantly higher than that of larger particles (Fig. S10a), thereby elevating the corresponding HBF525, RH levels during NPF events.

**Updates in the reference list:**

Andrews, E., Sheridan, P. J., Fiebig, M., McComiskey, A., Ogren, J. A., Arnott, P., Covert, D., Elleman, R., Gasparini, R., Collins, D., Jonsson, H., Schmid, B., and

Wang, J.: Comparison of methods for deriving aerosol asymmetry parameter, J. Geophys. Res. Atmos., 111, 1–16, https://doi.org/10.1029/2004JD005734, 2006.

Marshall, S. F., Covert, D. S., and Charlson, R. J.: Relationship between asymmetry parameter and hemispheric backscatter ratio: implications for climate forcing by aerosols, Appl. Opt., 34, 5–6, 1995.

**Comments in details:**

RC3. The authors divided the NPF into two classes, one is relatively polluted period and clean cases during heatwave-dominated period

Some sentences are really hard to follow. Such as "Heatwaves triggered NPF earlier and prolonged the subsequent growth." NPF events usually occurred in clean environment with low RH and high sunshine. What evidence does support NPF is triggered by heatwave?

Response: This can refer to our responses to RC1 and RC2 of Referee #2. The NPF events indeed started earlier and the duration of subsequent growth was longer during the heatwave-dominated P2 period, especially compared to the NPF events occurred during the P1 period and the same period in the summer of 2023 (Figure R4). Given that the formation mechanisms of different NPF events are out of the scope of this study, we have revised the corresponding expressions to highlight the differences in both NPF events and aerosol physicochemical properties under heatwave conditions.

**RC4. Line 34-36 This sentence is not supported by the data in Table S2 where the f(RH) is almost the same for NPF events and non-event days.**

**Response:** We thank the reviewer for pointing out this issue. The f(RH) values in the original Table S2 were rounded to one decimal place, which made the differences between event and non-event days negligible (but that is not the case, as shown in the below Figure R8). To better reflect the variations in aerosol optical and hygroscopic properties on NPF days and non-event days during different periods, we have revised Table S2 to include f(RH) and other relevant parameters with an additional decimal

place.

Figure R8. The box plots of f(RH) during P1 and P2 NPF days and non-event days

RC5. Line 90-93 NPF could alter the size distribution thereby aerosol optical properties, nonetheless, there is currently limited research on the impact of NPF on aerosol optical hygroscopicity (Ma et al., 2016; Ren et al., 2021). Since the particles with diameter less than 100nm have insignificant contribution to aerosol scattering just like showed in Figure S7.

Response: The influence of newly formed particles on the aerosol optical properties is insignificant according to the Mie theory, but previous studies have found that the subsequent growth of these new particles can enhance the hygroscopicity and extinction coefficient of aerosol populations (Cheung et al., 2020; Shen et al., 2011; Sun et al., 2024; Wu et al., 2015, 2116). We have revised the manuscript as follows:

L89-98: Numerous studies have demonstrated that f(RH) is influenced by the size distribution, in addition to particle chemical composition (Chen et al., 2014; Kuang et al., 2017; Petters and Kreidenweis, 2007; Quinn et al., 2005). There is currently limited research on the variations in aerosol optical hygroscopicity during NPF days despite significant changes in aerosol size distributions and chemical compositions, partly due to that newly formed particles insignificantly affect the optical properties of aerosols (Kuang et al., 2018). However, previous studies have observed the

enhancement in aerosol hygroscopicity (Cheung et al., 2020; Wu et al., 2015, 2016) and extinction coefficients (Shen et al., 2011; Sun et al., 2024) during the subsequent growth of NPF.

**Updates in the reference list:**

Shen, X. J., Sun, J. Y., Zhang, Y. M., Wehner, B., Nowak, A., Tuch, T., Zhang, X. C., Wang, T. T., Zhou, H. G., Zhang, X. L., Dong, F., Birmili, W., and Wiedensohler, A.: First long-term study of particle number size distributions and new particle formation events of regional aerosol in the North China Plain, Atmos. Chem. Phys., 11, 1565–1580, https://doi.org/10.5194/acp-11-1565-2011, 2011.

RC6. Line 142-144 As can be seen from Figure 1, the hourly temperature during P2 period (August 7-19) are not always above 40°C which in not consistent with these sentence. In addition, why do you choose the hourly total scattering coefficient at 525 nm of 100 Mm-1 as criteria?

**Response:** We thank the reviewer for pointing out these contradictions. As explained in our response to **RC1** of Referee #2, the standard for "heatwave-dominated" P2 period was updated accordingly in the main text (Figure R1). We chose 100 Mm-1 (i.e., the last  $10^{th}$  percentile level of all the measured  $\sigma_{sca, 525}$  records; Figure R1c) as the criteria for the relatively polluted condition of P1, and none of the hourly  $\sigma_{sca, 525}$  during the P2 period (i.e., within 16.1-94.4 Mm-1) exceeded this threshold.

We have revised the corresponding sentence:

**L281-284:** It should be noted that the hourly  $\sigma_{sca, 525}$  values during the P2 period were exclusively below 100 Mm-1 (approximately the last  $10^{th}$  percentile of  $\sigma_{sca, 525}$  data, regarded as the threshold value of relatively polluted cases; Fig. S2c), suggesting a much cleaner environment compared to the relatively polluted P1 period.

RC7. Line 171-173 What kind of assumption are behind this calculation of ALWC? What kind of data are used to estimate dry aerosol volume concentration ( $V_{dry}$ ) by a machine learning method?

**Response:** In this study, the method for calculating ALWC based on the measurements of humidified nephelometer system was proposed by Kuang et al. (2018). The method mainly consists of two steps, as shown in the flowchart in Figure R9 (redrawn from Figure 8 in Kuang et al., 2018).

**Figure R9.** The flowchart of calculating ALWC based on measurements of a three-wavelength humidified nephelometer system (Kuang et al., 2018).

First is the estimation of the dry aerosol volume concentration ( $V_{dry}$ ). According to the Mie theory, the aerosol  $\sigma_{sca, \lambda}$  is roughly proportional to  $V_{dry}$  (Pinnick et al., 1980). However, variations of the  $\sigma_{sca, \lambda}/V_{dry}$  ratio are largely influenced by PNSD, assuming that other factors affecting aerosol scattering efficiency (e.g., the refractive index and the BC mixing state) vary insignificantly. To derive a simple function describing the relationship between the measured  $\sigma_{sca, \lambda}$  and  $V_{dry}$ , a machine learning approach is utilized. The random forest model was trained and validated by using datasets of measured optical parameters including the dry  $\sigma_{sca, \lambda}$  and  $\sigma_{bsca, \lambda}$  at three wavelengths, which can reflect the variations in aerosol size distribution (e.g., the calculated HBF and SAE) (Kuang et al., 2018).

Secondly is the relationship between aerosol f(RH) and volume hygroscopic growth factor ( $f_V(RH)$ ). Based on the  $\kappa$ -Köhler theory (Petters and Kreidenweis, 2007), f(RH) can be parameterized by the aerosol optical hygroscopic parameter  $\kappa_{sca}$  (Brock et al., 2016). Assuming that the Kelvin effect on particles above 100 nm is negligible and that a constant hygroscopic parameter,  $\kappa$ , represents the overall hygroscopicity (Brock et al., 2016),  $f_V(RH)$  can be parameterized by the volume hygroscopic parameter,  $\kappa_V$  (Brock et al., 2016; Kuang et al., 2018). Therefore,  $\kappa_V$  is crucial for evaluating the wet volume of aerosols upon hygroscopic growth. Similarly, the variations of the ratio  $\kappa_V/\kappa_{sca}$  is largely influenced by PNSD (Kuang et al., 2018). Given that SAE can reflect the evolution of PNSD (Kuang et al., 2017, 2018) and assuming  $\kappa_{sca}$  reflects the overall hygroscopicity of aerosols, a lookup table reflecting  $\kappa_V/\kappa_{sca}$  can be constructed by inputting the measured SAE and  $\kappa_{sca}$  (Kuang et al., 2018). Consequently,  $f_V(RH)$  and the corresponding ALWC can be further evaluated.

We have added more details in the Supplement.

**L59-62:** where the dry aerosol volume concentration ( $V_{dry}$ ) was estimated with the dry scattering coefficient at three wavelengths utilizing a machine learning method (Kuang et al., 2018).

RC8. The authors mentioned the Nafion dryer are used to dry the ambient air in S1 section in the supplement, what's the total flow for online measurements? Both humidified nephelometers and SMPS share the same  $PM_{2.5}$  impactor? More information of SMPS should be given, such as the sheath flow and aerosol

flow, the sheath flow control mode, data retrieval, etc. In supplement S5, is the neutralizer model right?

**Response:** We have updated the details of the filed observation. We appreciate the reviewer's reminder, and the model of the soft X-ray neutralizer has been corrected (model 3088, TSI Inc.).

**In the main text:**

*L161-164:* Ambient air was firstly dried through a Nafion dryer (model MD-700, Perma Pure LLC) to ensure RH <35%, then split into two streams for both dry and

humidified nephelometers operated in parallel. The flowrate for each nephelometer was 2.6 LPM.

**In the Supplement:**

L36-39: The ambient air was sampled at a flowrate of 16.7 LPM through a  $PM_{2.5}$  impactor (model 2000-30EH, URG Inc.) and dried with a Nafion dryer (model MD-700, Perma Pure LLC), to achieve a low relative humidity level (RH <35%) prior to the online aerosol size distribution, optical and hygroscopic measurements.

**S5. Particle number size distribution measurements**

During the field observation, every 3-min PNSD and particle volume size distribution (PVSD) was measured by a SMPS, which consisted of a soft X-Ray neutralizer (model 3088, TSI Inc.), a differential mobility analyzer (model 3081, TSI Inc.), and a condensation particle counter (model 3775, TSI Inc.) (Dominick et al., 2018; Rissler et al., 2006). The SMPS was operated at a sheath/sample flow rate of 3.0/0.3 LPM, and the detected size range was 14.1-710.5 nm with 110 size bins. Data inversion of measured particle size distributions was achieved with the Aerosol Instrument Manager software (AIM, TSI Inc.), including the multiple charge and diffusion corrections (Denjean et al., 2015; Rosati et al., 2022).

**Updates in the reference list of Supplement:**

Denjean, C., Formenti, P., Picquet-Varrault, B., Camredon, M., Pangui, E., Zapf, P., Katrib, Y., Giorio, C., Tapparo, A., Temime-Roussel, B., Monod, A., Aumont, B., and Doussin, J. F.: Aging of secondary organic aerosol generated from the ozonolysis of α-pinene: Effects of ozone, light and temperature, Atmos. Chem. Phys., 15, 883–897, https://doi.org/10.5194/acp-15-883-2015, 2015.

Rosati, B., Isokääntä, S., Christiansen, S., Jensen, M. M., Moosakutty, S. P., De Jonge, R. W., Massling, A., Glasius, M., Elm, J., Virtanen, A., and Bilde, M.: Hygroscopicity and CCN potential of DMS-derived aerosol particles, Atmos. Chem. Phys., 22, 13449–13466, https://doi.org/10.5194/acp-22-13449-2022, 2022.

RC9. Line213-215 This sentence is not supported by the Figure S2, where the sum of the measured chemical composition mass concentration is higher than PM2.5 mass concentration. It is really hard to understand to use TSP results for the characterization of NPF.

**Response:** We agree on the concerns about the TSP chemical results used in this study, and we would like to clarify the rationale for their application despite potential limitations.

Previous studies on size-resolved aerosol chemical characterization have suggested that key components (e.g., sulfate, nitrate, and ammonium (SNA in short), OC, EC) of PM2.5 (or PM10) were predominantly concentrated in the submicron range (An et al., 2024; Bae et al., 2019; Chen et al., 2019; Duan et al., 2024; Kim et al., 2020; Xu et al., 2024). Specifically, SNA are the predominant fine-mode components (An et al., 2024; Bae et al., 2019; Chen et al., 2019; Kim et al., 2020; Xu et al., 2021, 2024), while organic compounds (OM) could exhibit a broader size distribution due to their diverse sources. For instance, primary organic aerosols (POA) are mainly concentrated in the accumulation mode, while secondary organic aerosols (SOA) may possess a relatively broader size distribution (Duan et al., 2024; Kim et al., 2020; Xu et al., 2021), depending on specific activities and emission sources (e.g., from boilers and kilns during industrial processes) (An et al., 2024). Nevertheless, such emissions with larger particle sizes were relatively limited at our mixed residential-commercial urban site (Chen et al., 2024). The discrepancy in the total mass concentration between the 24-h TSP samples and daily mean PM2.5 (of similar temporal variations; original Fig.S2) could be partly attributed to certain secondary organics and crustal elements (e.g., Ca2+, Mg2+); besides, the boxplot of hourly PM2.5 data actually spanned a wider range, which can generally cover the corresponding mass abundance of TSP samples despite some biases (Figure R10).

Figure R10. Mass concentrations of the measured chemical components for TSP filter samples, as well as the corresponding daily mean  $PM_{2.5}$  and  $PM_{10}$  results.

Given the lack of online aerosol chemical characterization on fine particles (e.g., PM2.5 or PM1) and that only the simultaneously collected TSP filter samples were available for offline chemical analysis (e.g., OC, EC, and water-soluble inorganic species), the obtained chemical composition results were utilized mainly for the investigation of aerosol/PM2.5 optical and hygroscopic properties in this study (since the mechanisms of different NPF events are out of the scope, as stated in our response to RC3 of Referee #3). The SOC/TOC ratio results derived from the TSP chemical composition data were mainly aimed to assess secondary formation ability and related impacts on aerosol optical and hygroscopic properties, rather than to explore the detailed mechanisms of NPF events. While the use of TSP samples contains some uncertainties, the bulk chemical information remains reasonable for our research objectives (i.e., the optical hygroscopic properties of PM2.5). Future studies of molecular-scale chemical characterization are needed to refine the analysis and deepen understanding on the role of chemical composition in both NPF events and aerosol physicochemical properties.

We have clarified the above points in the revised manuscript and updated accordingly on related figures as below:

L194-205: Results of the offline chemical analysis with TSP filter samples are provided in Sect. S3 and Fig. S3. It should be noted that certain secondary organics and crustal elements (e.g.,  $Ca^{2+}$ ,  $Mg^{2+}$ ) that could exhibit a broader size distribution may contribute to the observed discrepancy in the total mass concentration between the 24-h TSP samples and daily mean  $PM_{2.5}$  (of similar temporal variations; Fig.S3) (Duan et al., 2024; Kim et al., 2020; Xu et al., 2021). Nonetheless, previous studies reported that key components such as SNA (i.e.,  $SO_4^{2-}$ ,  $NO_3^{-}$ , and  $NH_4^{+}$ ) and primary organics of  $PM_{2.5}$  (or  $PM_{10}$ ) were predominantly concentrated within the submicron size range (An et al., 2024; Bae et al., 2019; Chen et al., 2019; Duan et al., 2024; Kim et al., 2020; Xu et al., 2024). While the use of TSP samples contains some uncertainties, the bulk chemical information remains reasonable for characterizing the optical and hygroscopic properties of  $PM_{2.5}$ .

**Updates in the reference list:**

Bae, M. S., Lee, T., Schauer, J. J., Park, G., Son, Y. B., Kim, K. H., Cho, S. S., Park, S. S., Park, K., and Shon, Z. H.: Chemical Characteristics of Size-Resolved Aerosols in Coastal Areas during KORUS-AQ Campaign; Comparison of Ion Neutralization Model, Asia-Pacific J. Atmos. Sci., 55, 387–399, https://doi.org/10.1007/s13143-018-00099-1, 2019.

Chen, Q., Mu, Z., Song, W., Wang, Y., Yang, Z., Zhang, L., and Zhang, Y. L.: Size-Resolved Characterization of the Chromophores in Atmospheric Particulate Matter From a Typical Coal-Burning City in China, J. Geophys. Res. Atmos., 124, 10546–10563, https://doi.org/10.1029/2019JD031149, 2019.

Duan, J., Huang, R. J., Wang, Y., Xu, W., Zhong, H., Lin, C., Huang, W., Gu, Y., Ovadnevaite, J., Ceburnis, D., and O'Dowd, C.: Measurement report: Size-resolved secondary organic aerosol formation modulated by aerosol water uptake in wintertime haze, Atmos. Chem. Phys., 24, 7687–7698, https://doi.org/10.5194/acp-24-7687-2024, 2024.

Kim, N., Yum, S. S., Park, M., Park, J. S., Shin, H. J., and Ahn, J. Y.: Hygroscopicity of urban aerosols and its link to size-resolved chemical composition during spring and summer in Seoul, Korea, Atmos. Chem. Phys., 20, 11245–11262, https://doi.org/10.5194/acp-20-11245-2020, 2020.

Xu, W., Chen, C., Qiu, Y., Xie, C., Chen, Y., Ma, N., Xu, W., Fu, P., Wang, Z., Pan, X., Zhu, J., Ngcg, N. L., and Sun, Y.: Size-resolved characterization of organic aerosol in the North China Plain: New insights from high resolution spectral analysis, Environ. Sci. Atmos., 1, 346–358, https://doi.org/10.1039/d1ea00025j, 2021.

Xu, W., Kuang, Y., Xu, W., Zhang, Z., Luo, B., Zhang, X., Tao, J., Qiao, H., Liu, L., and Sun, Y.: Hygroscopic growth and activation changed submicron aerosol composition and properties in the North China Plain, Atmos. Chem. Phys., 24, 9387–9399, https://doi.org/10.5194/acp-24-9387-2024, 2024.

**RC10. Line 228-229, Line 235-236 The two sentences are not consistent.**

**Response:** We have revised the sentences:

**L247-250:** This could be largely attributed to the reduction in anthropogenic emissions (e.g., NO2, CO, except SO2) from limited outdoor activities influenced by the heatwaves in P2, as well as partly suspended industries and transportation to alleviate the power shortage issue (Chen et al., 2024).

**RC11. Line 239-241 Are the mean values of $1.6 \pm 0.1$ and $1.7 \pm 0.2$ during the P1 and P2 periods different significantly?**

**Response:** We have conducted a Welch's t-test to evaluate the significance of the difference between the f(RH) values during different periods. The results indicate that the difference is statistically significant (p < 0.05). We have adjusted the corresponding sentence:

**L259-261:** The f(RH) was found to be relatively higher (p < 0.05) in heatwave days, with the mean values of  $1.61 \pm 0.12$  and  $1.71 \pm 0.15$  during the P1 and P2 periods, respectively.

**RC12. Line 241-243 Do you think the results is dependent on the algorithm of ALWC? Please clarify it.**

**Response:** The method of calculating ALWC by using the humidified nephelometer system has been presented in the response to **RC7** (Referee #3), which suggests that ALWC is largely dependent on the difference in aerosol volume concentration (e.g., related with the aerosol loading) between the dry and humidified conditions. We have revised it as below:

L261-266: Differently, ALWC was more abundant during the normally hot P1 period than the heatwave-dominated P2 period. This is likely due to that the derivation algorithm of ALWC utilized in this study (Kuang et al., 2018) was partly dependent on (e.g., positively correlated) the dry aerosol scattering coefficient, or rather the aerosol volume concentration in the dry condition (refer to Sect. S3 and Fig. S11 of the supplement).

**RC13. Line 278-279 relatively polluted?**

**RC1** of Referee#2 (Figure R1). The mean  $\sigma_{sca, 525}$  and PM2.5 during P1 period were 113.6% and 49.5% higher than those for P2 period, respectively, and the mean visibility was 22.1% lower than that of P2 period. In addition, the mean CS of P1 NPFP events is higher than 0.015 s-1, which can be also identified as the polluted-type NPF day (Shang et al., 2023). We have revised the manuscript accordingly.

**L310-315:** As stated in Sect.3.1, NPF events during the P1 period tended to occur in relatively polluted environments compared to that of P2 NPFC, HW events, as evidenced by the frequent occurrence of  $\sigma_{sca, 525} > 100 \text{ Mm}^{-1}$ , increased air pollutant concentrations and lower visibility levels during P1 (Table S2, Fig. 1). Additionally,

the mean CS of the  $NPF_P$  events was above 0.015 s-1 (Table S2), which could be considered as the "polluted" NPF day (Shang et al., 2023).

**Updates in the reference list:**

Shang, D., Hu, M., Tang, L., Fang, X., Liu, Y., Wu, Y., Du, Z., Cai, X., Wu, Z., Lou, S., Hallquist, M., Guo, S., and Zhang, Y.: Significant effects of transport on nanoparticles during new particle formation events in the atmosphere of Beijing, Particulogy, 80, 1–10, https://doi.org/10.1016/j.partic.2022.12.006, 2023.

**RC14. Line 284-285 the upper detection limit of 30 km? please clarify it**

**Response:** Although no specific information on the upper limit of the visibility data used in this study (i.e., from the Integrated Surface Database (ISD)), long-term observations using the same visibility records consistently show values not exceeding 30 km (Mukherjee and Toohey, 2016; Xu et al., 2020; Zhao et al., 2022). To avoid unnecessary misleading, we have revised the sentence:

**L317-319:** In addition, the mean  $PM_{2.5}$  concentration was even lower than  $10.0 \,\mu\text{g}\cdot\text{m}^{-3}$ , and the corresponding visibility level was almost maintained at 30 km (Fig. 1e).

RC15. Line 291-292 why the authors emphasize that "sulfuric acid concentration was a critical factor for the occurrence of P1 NPF events."? Do you mean P1 NPF and P2 NPF different? Figure2f show the diurnal variation of H2SO4 during different periods with minor difference.

**Response:** We thank the reviewer for raising this important question. NPFP events typically occurred around 11:00 LT during the P1 period, and the H2SO4 (SO2) concentration on NPFP days was 5.3% (19.3%) higher than that on non-event days at this time. Although NPFC, HW events occurred approximately one hour earlier during the P2 heatwave period, the corresponding H2SO4 (SO2) concentration on non-event days was conversely higher. This likely suggests a minor role of H2SO4 (SO2) concentration in the P2 NPFC, HW events. As discussed in the responses to **RC10** (Referee #2) and **RC17** (Referee #3), the extremely high temperatures likely

suppressed nucleation and growth processes on the P2 non-event days, even when H2SO4 (SO2) concentrations were elevated.

Moreover, the predominant precursor species responsible for NPF events may vary under heatwave weather, depending on the concentration/type of precursors in the environment (Ma et al., 2016; Wang et al., 2017). For instance, recent studies have found that heatwaves likely led to changes in the abundance of precursors such as VOCs from both anthropogenic and biogenic sources (Chen et al., 2024).

Given the above reasons, we suggest that sulfuric acid could be a more critical factor for the occurrence of NPFP events during the P1 period, in comparison to the P2 heatwayes.

RC16. Line 296-297 This might suggest that meteorological factors might not be the predominant determining factor of NPF occurrence? In this study, the measurement period is so short, more caution should be paid to reach this conclusion.

**Response:** We appreciate the reviewer for the comment and have adjusted the expression to avoid arbitrary statement:

L330-332: This likely suggests that meteorological factors might not be the predominant determining factor of NPF occurrence during the heatwaves of 2022 summer in urban Chongqing,

RC17. Line 309-311 "...the occurrence and subsequent growth of NPF during non-event days...", the sentence should be clarified.

**Response:** This can refer to our response to **RC10** of Referee #2. The higher temperatures likely prevented nucleation processes from meeting the criteria for NPF events and hindered the subsequent growth of nucleation mode particles, even if the concentrations of SO2 and H2SO4 were higher on the P2 non-event days. We have revised the sentence:

**L345-348:** Hence, the even higher temperature (e.g., T > 40 °C) likely suppressed the nucleation processes and the subsequent growth of nucleation mode particles on P2

**RC18. Line 319 NPF could occur worldwide, what's the temperature threshold of NPF events?**

**Response:** The occurrence of NPF depends on a complex interplay of factors, including precursor gas types/concentrations, meteorological conditions, and pre-existing aerosol loading (Kerminen et al., 2018; Kulmala et al., 2003). Hence, there is no specifically defined temperature threshold of NPF events, although which have been observed around the world throughout the years (Crumeyrolle; et al., 2023; Dada et al., 2017). Accordingly, we have revised the content:

L357-361: Since the sunrise and sunset time did not significantly vary within the study period (i.e., less than a half hour discrepancy), heatwaves likely provided more favorable conditions (e.g., enhanced volatile gaseous emissions, low RH; Bousiotis et al., 2021; Hamed et al., 2007; Wang et al., 2024) for the occurrence of NPF events in urban Chongqing.

**Update in the reference list:**

Hamed, A., Joutsensaari, J., Mikkonen, S., Sogacheva, L., Dal Maso, M., Kulmala, M., Cavalli, F., Fuzzi, S., Facchini, M. C., Decesari, S., Mircea, M., Lehtinen, K. E. J., and Laaksonen, A.: Nucleation and growth of new particles in Po Valley, Italy, Atmos. Chem. Phys., 7, 355–376, https://doi.org/10.5194/acp-7-355-2007, 2007.

**RC19. Line 359-360 during P2 heatwave-dominated NPF events? The meaning is not clear.**

**Response:** We have revised the corresponding expression:

**L405-407:** Both HBF and SAE on P2 NPFC, HW days were significantly higher than that of P1 NPFP cases (Fig. 3c, e), largely due to the smaller  $R_{eff}$  observed during heatwave-dominated period (Table S2).

**RC20. The PVSDs in Figure S3 a2-c2 are strange above ~ 500 nm, why?**

**Response:** Upon checking the original size distribution data, we found that this phenomenon could be attributed to the following two factors. Firstly, the number concentrations of particles above 500 nm are quite close to each other and generally lower than 500 #/cm³, while these large particles can significantly contribute to the particle volume concentration. Secondly, the SMPS has larger size intervals (i.e., fewer size bins) for particle sizes above 500 nm compared to smaller size ranges. Besides, a similar pattern was observed for the PVSDs above 500 nm during P2 non-event days (Figure R11). However, occasionally occurred extremely high volume concentrations could render this effect comparatively less noticeable.

Figure R11. The PVSDs (a2-d2) of particles above 500 nm on P1 NPF event (a2) and non-event (b2) days, P2 NPF event (c2) and non-event (d2) days.

**RC21. Section 3.3 Characteristics of the aerosol optical and hygroscopic properties during NPF events?**

Line 338-339 What the meaning of NPF events in this study? Refer to the whole day?

**Response:** As clarified in our response to **RC1** of Referee #3, we have updated the section title to "3.3 Characteristics of the aerosol optical and hygroscopic properties on different types of NPF days". The mentioned time periods with "NPF events" have

been replaced with "NPF days", i.e., the whole day when NPF was observed, throughout the manuscript.

RC22. Line 403-405 This opinion could not supported by the data in Table S2, where the fw during P1 and P2 NPF are  $0.47 \pm 0.04$ ,  $0.48 \pm 0.05$ , while  $0.46 \pm 0.04$  and  $0.46 \pm 0.06$  during P1 and P2 non-event days.

**Response:** We agree that the discrepancy in the corresponding mean  $f_W$  results is relatively insignificant, while the differences in the diurnal variations of  $f_W$  between NPF and non-event days were more pronounced. We have modified the content to highlight that during the subsequent growth and aging of pre-existing and new particles, the  $f_W$  of NPF days was higher than that of non-event days in the afternoon: **L448-453:** The  $f_W$  levels were slightly higher during NPF days in comparison to that of non-event days (Table S2). This difference was more pronounced in the afternoon of NPF days (e.g., even exceeded 50%; Fig. 3f), verified the enhancement of aerosol hygroscopicity during the subsequent growth and atmospheric aging of both pre-existing and newly formed particles.

RC23. Line 407-410, The authors mentioned "data mainly within the time window of 08:00-22:00 were utilized for the following discussion", but, Figure 4a and 4b included other data, please clarify it.

**Response:** The full diurnal patterns of NFAcc. and VFAcc. in Figure 4a and 4b were shown to illustrate the significant impact of NPF events and subsequent growth on aerosol size distributions, particularly in comparison to non-event days. We have highlighted the time window of 08:00-22:00 LT in Figure 4a and 4b to facilitate the corresponding discussion (see in **RC5** of Referee #2) and updated the manuscript accordingly.

**L481-484:** This is mainly due to the explosive formation of ultrafine particles and subsequent growth on NPF days, significantly altering aerosol size distributions and inducing large fluctuations in the number and volume fractions of accumulation mode particles (as shaded in Fig. 4a-b).

**RC24. Why only a few data in Figure 4d is with the temperature above 40°C**

**Response:** In fact, the high values of f(RH) (e.g., f(RH) > 1.7) generally corresponded to extremely high temperatures above 40 °C on P2 non-event days (see Figure R12 as below). During the heatwave-dominated P2 period, intensified photochemical reactions led to the formation of more hygroscopic secondary aerosols, which increased both  $R_{eff}$  and f(RH).

The few data points highlighted by the red dashed circle in Figure 4d were attempted to emphasize that even smaller particles ( $R_{\rm eff}$  <110 nm) could exhibit high f(RH) levels under such extreme temperature conditions (T >40 °C). This likely highlights the significant impact of intense photochemical reactions on aerosol hygroscopicity during the heatwaves, particularly in comparison to P1 non-event days.

Figure R12. The relationship of f(RH) with  $R_{eff}$  and temperature (as indicated by the colored dots) on P2 non-event days.

RC25. Figure 5 Why the authors labeled polluted and relatively polluted in Figure 5a and 5b for P1 NPF and P2 NPF, respectively? which is not consistent "in clean environment" mentioned above in the manuscript?

**Response:** We appreciate the reviewer's attention to this inconsistency. In the original version, we intended to indicate that lower values of the SAE and f(RH) corresponded to higher scattering coefficients, reflecting higher aerosol loading conditions. To avoid unnecessary misunderstandings, we have removed the "polluted" and "relatively polluted" labels from Figure 5.

RC26. Line 447-449 Such a positive (negative) correlation of f(RH) with SAE (CS) was more pronounced in heatwave-induced high temperature days during P2 period. Which is not supported by the correlation R = 0.58 during P2 NPF in Figure 5b1, while R = 0.65 during P1 NPF in Figure 5a1.

**Response:** Thanks for pointing out this mistake. Upon re-examining the original data, we confirmed that the correlation coefficient between f(RH) and SAE during P1 NPFP days (R = 0.65) is indeed higher than that during P2 NPFC, HW days (R = 0.58). This may be attributed to the different NPF events during the P2 heatwave-dominated period, as detailed in our response to **RC2** (Referee #2). We have removed such erroneous conclusions from both Section 3.4 and the final conclusions section:

**L498-500:** Aerosol f(RH) and SAE exhibited a higher level on P2  $NPF_{C, HW}$  days (as shown by the dash lines in Fig. 5), the possible reasons can be attributed to the following two aspects.

**L671-673:** A significantly positive (negative) correlation between f(RH) and SAE (CS,  $\sigma_{sca, 525}$ , or rather the pollution level) was observed on NPF days for both periods, accompanied by higher f(RH) and SAE values on NPFC, HW days.

RC27. Line 463-465 It is worth noting that f(RH) did not show a consistently higher level after the NPF occurrence during P2 period, and it was slightly higher within the first few hours of NPF occurrence during P1 NPF events (Fig. 3b). Which is hard to see from Figure 3b.

**Response:** As shown by the dashed circle in Figure R13, the f(RH) on P1 NPFP days was higher than that on P2 NPFC, HW days during the time period of ~ 12:00-15:00 LT (i.e., following the NPF occurrence).

*Figure R13.* Diurnal variation f(RH) on NPF days during P1 (red line) and P2 (blue line) periods. The shaded areas stand for the corresponding  $\pm 1\sigma$  standard deviations.

We have included the specific time period in the revised manuscript:

**L514-515:** and it was slightly higher within the first few hours of NPF occurrence (i.e.,  $\sim 12:00$ -15:00 LT) on P1 NPFP days (Fig. 3b).

**L683-685:** further leading to a lower f(RH) following the NPF occurrence (i.e.,  $\sim$  12:00 -15:00 LT) in comparison to P1 NPFP days.

RC28. Line 474-476 The critical sizes corresponding to the cumulative frequency of 50% in  $\sigma_{sca, 525}$  were 358.7 nm and 333.8 nm on P1 and P2 NPF event days, respectively. Have you seen the particles grow to this particles during NPF events?

**Response:** The aerosol optical properties (e.g., light scattering) are inherent and influenced by their size distributions. The total aerosol scattering coefficient is predominantly influenced by larger particles, which are unnecessarily originate solely from the growth of newly formed particles. Instead, these larger particles could be resulted from the mixing of pre-existing particles with newly formed ones that may undergo subsequent growth and aging processes. Therefore, the observed critical sizes corresponding to the cumulative frequency of 50% in  $\sigma_{sca, 525}$  (D50) likely represent a combination of pre-existing and aged larger particles. Additionally, similar D50 values were observed on non-event days, indicating that such size ranges are not exclusive to

NPF events. We have revised the sentences to highlight the contribution of the pre-existing and aged large particles:

**L526-529:** This indicates that relatively smaller particles including the newly formed and grown ones mixed with pre-existing and aged particles contributed a slightly higher portion to  $\sigma_{sca, 525}$  on P2 NPFC, HW days, while the  $\sigma_{sca, 525}$  was mainly contributed by larger ones on P1 NPFP days.

**RC29. Line 485-486 "...leading to a reduced enhancement in aerosol light scattering...", please make it clear**

**Response:** We have revised the sentence as below:

**L534-537:** Newly formed ultrafine particles contributed minor to aerosol optical properties, resulting in a lower f(RH) during the initial hours of P2 NPFC, HW events compared to that of P1 NPFP events (Fig. 3b), as evidenced by a smaller  $R_{eff}$  for P2 NPFC, HW events (Fig. S6).

RC30. Line 540-542 It should be noted that the reported  $f_{RF}(RH)$  for the UGR site (Spain) was even higher, likely due to the relatively larger HBF in that area (Titos et al., 2014; 2021). This is not supported by the data in black dots (black dots for urban sites, UGR is an Urban site) Figure 2 in Titos et al 2021, although it is really hard to see which black dot is for UGR.

**Response:** Upon re-examining the raw HBF data in Titos et al. (2014), we acknowledge that attributing the higher  $f_{RF}(RH)$  in the UGR site to a larger HBF may not be fully justified, as the mean HBF (0.15) for UGR are comparable to that during P2 period (0.153) in this study. To clarify this discrepancy, we identified a key methodological difference: Titos et al. (2014) originally used a surface reflectance constant (RS) of 0.15, consistent with the globally averaged value adopted in our and previous studies (Fierz-Schmidhauser et al., 2010; Xia et al., 2023; Zhang et al., 2015). However, in their 2021 study, the RS was changed to 0.25 (for rural, urban and mountain sites), which likely contributed to the higher  $f_{RF}(RH)$  values reported for the UGR site. If the constant RS = 0.25 was used in the derivation of  $f_{RF}(RH)$  in this study,

the mean  $f_{RF}(RH)$  (nearly 2.5) on the P2 NPFC, HW days would be higher than that in the UGR site. Other factors such as variations in the mass scattering efficiency ( $\alpha_s$ ) could also contribute to the observed differences. As Titos et al. (2020) mentioned, aerosol scattering was largely enhanced after water uptake in URG, suggesting that  $\alpha_s$  was likely higher in the UGR site under the condition of similar f(RH) levels between the two sites.

We have revised the content in the manuscript:

**L600-602:** It should be noted that the reported  $f_{RF}(RH)$  for the UGR site (Spain) was even higher, likely due to the higher  $R_s$  and  $\alpha_s$  used in the derivation of  $f_{RF}(RH)$  in that area (Titos et al., 2021).

RC31. Line 565-566 "the new particles of higher hygroscopicity could contribute more to the activation of CCN," this opinion is not supported by aerosol optical hygroscopicity measurement, however, could be supported by HTDMA hygroscopicity measurement.

**Response:** Thanks for the suggestion, and we have revised the content accordingly:

L624-629: On the other hand, a large number of studies have demonstrated that the new particles of higher hygroscopicity could contribute more to the activation of CCN (Ma et al., 2016; Ren et al., 2021; Rosati et al., 2022; Sun et al., 2024; Wu et al., 2015), thereby modulating the aerosol-cloud interactions and further the global climate (Fan et al., 2016; Merikanto et al., 2006; Westervelt et al., 2013).

**Updates in the reference list:**

Fan, J., Wang, Y., Rosenfeld, D., and Liu, X.: Review of aerosol-cloud interactions: Mechanisms, significance, and challenges, J. Atmos. Sci., 73, 4221–4252, https://doi.org/10.1175/JAS-D-16-0037.1, 2016.

Merikanto, J., Spracklen, D. V, Mann, G. W., Pickering, S. J., and Carslaw, K. S.: Atmospheric Chemistry and Physics Impact of nucleation on global CCN, Atmos. Chem. Phys, 9, 8601–8616, 2009.

Rosati, B., Isokääntä, S., Christiansen, S., Jensen, M. M., Moosakutty, S. P., De Jonge,

R. W., Massling, A., Glasius, M., Elm, J., Virtanen, A., and Bilde, M.: Hygroscopicity and CCN potential of DMS-derived aerosol particles, Atmos. Chem. Phys., 22, 13449–13466, https://doi.org/10.5194/acp-22-13449-2022, 2022.

Westervelt, D. M., Pierce, J. R., Riipinen, I., Trivitayanurak, W., Hamed, A., Kulmala, M., Laaksonen, A., Decesari, S., and Adams, P. J.: Formation and growth of nucleated particles into cloud condensation nuclei: Model-measurement comparison, Atmos. Chem. Phys., 13, 7645–7663, https://doi.org/10.5194/acp-13-7645-2013, 2013.

**RC32. Figure S8, It seems the fitting equation wrong.**

**Response:** Thank you for pointing out this mistake, and we have corrected the fitting equation.

Figure S11. Correlation between the particle volume concentration determined by SMPS and  $\sigma_{sca, 525}$  measured by the humidified nephelometer system during the study period. The solid line represents the fitting line.

**Reference**

Mukherjee, A. and Toohey, D. W.: A study of aerosol properties based on observations of particulate matter from the U.S. Embassy in Beijing, China, Earth's Futur., 4, 381–395, https://doi.org/10.1002/2016EF000367, 2016.

Pinnick, R. G., Jennings, S. G., and Ch, P.: Relationships Between Extinction, Absorption, Backscattering, and Mass Content of Sulfuric Acid Aerosols, J. Geophys. Res., 85, 4059–4066, 1980.

Titos, G., Lyamani, H., Cazorla, A., Sorribas, M., Foyo-Moreno, I., Wiedensohler, A., and Alados-Arboledas, L.: Study of the relative humidity dependence of aerosol light-scattering in southern Spain, Tellus, Ser. B Chem. Phys. Meteorol., 66, https://doi.org/10.3402/tellusb.v66.24536, 2014.

Xu, W., Kuang, Y., Bian, Y., Liu, L., Li, F., Wang, Y., Xue, B., Luo, B., Huang, S., Yuan, B., Zhao, P., and Shao, M.: Current Challenges in Visibility Improvement in Southern China, Environ. Sci. Technol. Lett., 7, 395–401, https://doi.org/10.1021/acs.estlett.0c00274, 2020.

Zhao, G., Hu, M., Zhang, Z., Tang, L., Shang, D., Ren, J., Meng, X., Zhang, Y., Feng, M., Luo, Y., Yang, S., Tan, Q., Song, D., Guo, S., Wu, Z., Zeng, L., Zhang, Y., and Xie, S.: Current Challenges in Visibility Improvement in Sichuan Basin, Geophys. Res. Lett., 49, https://doi.org/10.1029/2022GL098836, 2022.

---

## Author Response (AR2)

Dear Editor,

We thank for all the constructive comments and suggestions from referees. We have carefully addressed and provided detailed explanations for their concerns. Point-by-point responses to the suggestions, corresponding updates with the revised manuscript, and the finalized version have been uploaded.

In the following, original suggestions, our response, and updates on the revised manuscript are shown in **bold**, normal, and *italic*, respectively.

Kind Regards,

Jing Chen, Yuhang Hao, and Peizhao Li

**Anonymous Referee #2**

**General comments:**

The authors have addressed the comments and suggestions satisfactorily, which has raised some additional questions and clarifications, as noted below. These should be addressed before the manuscript is accepted for publication.

**Response:** Thanks for the comments and suggestions.

**Specific comments:**

RC1. It is now clear that the NPF frequencies in 2022 (heatwave) and 2023 (no heatwave) were identical. Furthermore, the NPF frequencies were similar between P1 and P2 in 2022 (study period). This indicates that NPF was not triggered or induced by the extremely hot conditions (heatwave) in 2022. The NPF start times in 2022 and 2023 are also comparable (±1 hour). If the mean NPF start time is calculated, it is likely to match closely. Therefore, the authors should modify the manuscript title accordingly, replacing "induced by heatwaves" with "during heatwaves" or "during extremely hot conditions in 2022." As a result, the nomenclature used is also not appropriate. I again suggest changing it to NPF polluted and NPF clean. HW

**Response:** We appreciate this suggestion. We have adjusted the title into "Divergent changes in aerosol optical hygroscopicity and new particle formation during heatwaves of summer 2022", as well as the nomenclatures into " $NPF_{polluted}$ " and " $NPF_{clean, HW}$ " throughout the manuscript/supplement including figures.

RC2. P2 refers to clean (HW) time period with lower PM2.5 (or aerosol scattering coefficient) which means that lower scavenging loss for smaller particles, however, I see that lower-size particles are consistently missing during this time period (Fig. 1i), while the banana starts from the lowest measured size during P1. How do you explain this? This is clearly obvious from the contour plot of PNSDs (Fig. 1i)

Response: We have confirmed the reliablity of field observations first by checking through the raw size distribution data. The reduced concentrations of nucleation mode particles during P2 are likely attributed to the influence of transport on the nucleation process during heatwaves (Cai et al., 2023; Lee et al., 2019). Namely, some nucleation mode particles were transported from upwind regions and had undergone atmospheric aging thereby a certain degree of growth upon arrival, resulting in relatively lower concentrations of smaller-sized particles than the case of locally formed. Such phenomena have also been documented in previous studies. For instance, some NPF events observed on Fukue Island exhibited particle growth without high concentrations of sub-5 nm particles, a possible explanation for which is that nucleation occurred upstream of the observation site, with newly formed particles growing to larger sizes during transport (Lee et al., 2019). Similarly, Cai et al. (2023) concluded that regional transport played a critical role during NPF events in the Lulang River valley, where the concentrations of 5-10 nm particles were consistently lower than those of 10-50 nm particles.

Here we selected two different NPF days to demonstrate the cumulative contributions of local growth and transportation to the number concentration of nucleation mode particles (Figure R1). The calculation methods for the contributions of local growth and transport factor were introduced in Cai et al. (2018). The nearly negative contribution of transportation implies that the NPFpolluted events in P1 were mainly driven by local growth (Figure R1a), while transportation had outweighted local growth as a key factor for P2 NPFclean, HW days during heatwaves. Nevertheless, the comprehensive investigation of the mechanisms driving different NPF events, including the contribution of transportation to NPF, will be detailed in another work and is out the scope of this study. We have revised the manuscript as below to address the concern:

L358-363: The reduced  $N_{Nuc.}$  during P2 period were likely attributed to the influence of transport on the local nucleation process (Fig. S4; Cai et al., 2023; Lee et al., 2019). Namely, some nucleation mode particles transported from upwind regions had undergone atmospheric aging thereby a certain degree of growth upon arrival (Cai et

al., 2023), resulting in relatively lower concentrations of smaller-sized particles than the case of locally formed.

**Updates in the reference list:**

Cai, R., Chandra, I., Yang, D., Yao, L., Fu, Y., Li, X., Lu, Y., Luo, L., Hao, J., Ma, Y., Wang, L., Zheng, J., Seto, T., and Jiang, J.: Estimating the influence of transport on aerosol size distributions during new particle formation events, Atmos. Chem. Phys., 18, 16587–16599, https://doi.org/10.5194/acp-18-16587-2018, 2018.

Lee, K., Chandra, I., Seto, T., Inomata, Y., Hayashi, M., Takami, A., Yoshino, A., and Otani, Y.: Aerial observation of atmospheric nanoparticles on Fukue Island, Japan, Aerosol Air Qual. Res., 19, 981–994, https://doi.org/10.4209/aaqr.2018.03.0077, 2019.

Figure R1. The cumulative contributions of transportation and local growth to the nucleation mode particle number concentration on a  $NPF_{polluted}$  day (a) and a  $NPF_{clean, HW}$  day (b) of this study.

RC3. The air mass history clearly shows that the origin of the air masses was the same across different event types. However, as the trajectories are overlaid, not all of them are clearly visible. The authors could examine the air mass history over three days and also consider the trajectory altitude to determine if there are any differences.

**Response:** Following the reviewer's suggestion, we have examined the 48-h and 72-h

air mass history back trajectories at 500 or 1000 m altitude above the site during the study period (Figure R1). The results consistently show that air masses during the study period predominantly originated from the southern region, regardless of event types. This homogeneity suggests that air mass origin was unlikely a primary driver of the observed differences in NPF event characteristics. We have replaced the original Figure S4b into the Figure R2.

Figure R2. The 48-h and 72-h air-mass back trajectories at 500 or 1000 m altitude during the study period.

**Updated on the Supplement:**

L133-137: By using the HYSPLIT (Hybrid Single-Particle Lagrangian Integrated Trajectory) 4 model developed by NOAA (Stein et al., 2015), the 48-h and 72-h back trajectories of air masses at 500 or 1000 m altitude above the observation site during this study period were calculated and visualized by MeteoInfoMap (version 3.9.9; Figure S4b) (Chen et al., 2021; Tian et al., 2021; Wang, 2014).

**RC4. The NPF frequencies in Fig. S4 differ from those presented in Fig. R2(a).**

Response: Thank you for pointing out the inconsistency due to our carelessness.

Upon re-examining the original data, we identified a missing day of size distribution measurements during P1. This day was excluded as a null value in the revised analysis, but we forgot to update Fig. R2(a) in the initial response. We have now corrected it accordingly.

**Figure R3.** The occurrence frequencies of NPF, non-event and Undefined days during P1 ( $P1^{2023}$ ) and P2 ( $P2^{2023}$ ) periods of summer 2022 (2023).

RC5. What does NFACC, stand for? Does it refer to the number fraction of accumulation mode particles? Why focus on accumulation mode and number fractions? It would be more appropriate to first examine nucleation and Aitken mode particle number (absolute) concentrations, and then number fractions may be presented.

**Response:** Yes, the NFAcc. (VFAcc.) stands for the number (volume) fraction of accumulation mode particles in this study. According to reviewer's suggestion, we have displayed the number (volume) concentrations of different mode particles in Figure R4. Despite their lower number concentration compared to ultrafine particles, the accumulation mode particles contribute significantly to aerosol volume concentrations and light scattering (Figure R4, Figure S10a). Therefore, it is more appropriate to correlate NFAcc. and VFAcc. with the aerosol optical and hygroscopic properties (Figure 4).

We have revised the manuscript as follows:

L353-358: Distinct particle size distributions were observed for different NPF event

days. While the number concentrations of Aitken mode particles ( $N_{Ait}$ ) were comparable during NPF days of both periods, the corresponding number concentration of nucleation mode ( $N_{Nuc}$ ) was significantly higher on P1 NPFpolluted days (1880.8  $\pm$  2261.5 cm-3) than that for P2 NPF cases (1132.0  $\pm$  1333.5 cm-3) (Fig. 1i, Fig. S7).

L422-423: Given that NPF would largely enhance the abundance of both nucleation and Aitken mode aerosols (Fig. S7),

**L469-472:** Although ultrafine particles exhibited higher number concentrations during the study period, accumulation mode particles dominated the aerosol volume concentration and consequently contributed predominantly to the total light scattering (Figs. S7, S13).

**L618-621:** characterized with the smallest aerosol  $R_{eff}$  (102.8  $\pm$  12.4 nm) (Figure. S6), lowest number concentration (1897.0  $\pm$  680.8 cm-3) and fraction (0.20  $\pm$  0.10) of accumulation mode particles, intensified photooxidation, and a higher SOC/OC ratio.

Updated on the Supplement:

The Figure R4 has also been added into the Supplement (i.e., Figure S7):

**L150-153:** The PNSD is typically categorized into three modes: the nucleation mode  $(D_p < 25 \text{ nm})$ , Aitken mode (25-100 nm), and accumulation mode  $(D_p > 100 \text{ nm})$  (Zhu et al., 2021). The number concentrations and volume concentrations of different mode particles for different event categories are shown in Figure S7.

**Figure R4.** The number concentrations (left column) and volume concentrations (right column) of different mode particles for different event categories.

RC6. In Figs. 4 and 5, it is unclear how SAE and CS can be comparable. SAE is wavelength-dependent parameter and represents the total aerosol present in the atmosphere, whereas CS is calculated based on the measured size distribution within a specific size range (10–700 nm).

**Response:** We agree with this insightful comment. While our original intention was to highlight that higher SAE generally corresponded with lower scattering coefficients and cleaner conditions that is more favorable for NPF (lower CS), we acknowledge that SAE is not directly connected with CS. To avoid unnecessary misleading, we have removed the corresponding information of CS from Figure 5 and related discussions from the manuscript.

**L506-510:** Given that larger  $\sigma_{sca, 525}$  values typically indicate the condition of a higher aerosol loading, f(RH) increased with SAE whereas decreased with  $\sigma_{sca, 525}$ , or rather the pollution level, during NPF days. The cleaner environment of P2 period may

further favor the occurrence of NPF event.

**L701-703:** A significantly positive (negative) correlation between f(RH) and SAE ( $\sigma_{sca, 525}$ , or rather the pollution level) was observed on NPF days for both periods, accompanied by higher f(RH) and SAE values on  $NPF_{clean, HW}$  days.

Figure 5. The relationship between f(RH) and  $SAE_{635/450}$ , as well as temperature (as indicated by the color of dots, missing values are represented in gray), on P1 non-event days (a1), NPFpolluted days (a2) during the 08:00-22:00 LT time window. The vertical (horizontal) dash line represents the median value of  $SAE_{635/450}$  (f(RH)). (a3) The corresponding  $\sigma_{sca, 525}$  under different  $SAE_{635/450}$  levels on P1 NPFpolluted days. (b1-b3) The same but for P2 period.

**RC7. What is the average FR and GR for P1 and P2? Are they similar, or different?**

**Response:** We have added the mean FR and GR results into the Table S2 into the Supplement, and both FR and GR were generally lower for P2 NPFclean, HW days. We

have also updated the corresponding results in the manuscript. Please kindly see Table S2 and our original response (RC6 of Referee #2).

**Anonymous Referee #3**

The authors have revised the manuscripts thoroughly. One major concern on the discrepancy observed in the ratio of HBF525, RH/HBF525 with previous studies was left.

**Response:** In our study, the cases of HBF525, RH/HBF525 >1 were generally observed under conditions of extremely low  $\sigma_{sca, 525}$  and  $\sigma_{bsca, 525}$  levels (Figure R5), particularly during NPF event days in the P2 heatwave period. This signifies that contributions of optically-insensitive ultrafine particles to  $\sigma_{sca, 525}$  and especially to  $\sigma_{bsca, 525}$  were amplified upon humidification, leading to significantly higher HBF of these hydrated fine particles thereby elevated HBF525, RH levels (i.e., HBF525, RH/HBF525 >1).

**Figure R5.** The relationship between f(RH),  $f_b(RH)$ ,  $\sigma_{sca, 525}$  (symbol color),  $\sigma_{bsca, 525}$  (symbol size) in this study.

The higher HBF525, RH/HBF525 ratios observed in this study fundamentally reflect that the enhancement of aerosol backscattering coefficient ( $\sigma_{bsca}$ ) upon hygroscopic growth can be stronger than that of the total light scattering ( $\sigma_{sca}$ ) (Eq. 1). Namely, the  $f_b(RH)$  (aerosol backscattering enhancement factor) was higher than f(RH).

$$HBF_{525, RH}/HBF_{525} = \frac{\sigma_{bsca525, RH}}{\sigma_{sca525, RH}} / \frac{\sigma_{bsca,525}}{\sigma_{sca,525}} = \frac{\sigma_{bsca525, RH}}{\sigma_{bsca,525}} / \frac{\sigma_{sca525, RH}}{\sigma_{sca,525}} = \frac{f_b(RH)/f(RH)}{f(RH)}$$
 (1)

While the observed HBF525, RH ( $f_b(RH)$ ) generally exceeding HBF525 (f(RH)) seems to be different from previous studies (Titos et al. 2021; Xia et al. 2023), this phenomenon is actually not unprecedent (Figure R6). For instance, the higher HBF525, RH (or higher  $f_b(RH)$ ) could be found in Figure 2 of Fierz-Schmidhauser et al. (2010) and Figures 6-8 of Carrico et al. (2003) (Figure R6). As a matter of fact, this concern can be partly addressed theoretically with the Mie model.

Figure R6. Scatterplot of  $f_b(RH)$  versus f(RH) in this study (the mean levels for both P1 and P2 periods, and for different event categories), results from the study of Xia et al. (2023), Fierz-Schmidhauser et al. (2010) and Carrico et al. (2003). The error bars stand for the corresponding  $\pm$  one standard deviations.

Namely, the size-dependent efficiencies of light scattering ( $\sigma_{sca}$ ), backscattering ( $\sigma_{bsca}$ ) and hemispheric backscattering fraction (HBF= $\sigma_{bsca}/\sigma_{sca}$ ) in dry conditions, as well as the corresponding enhancements in these efficiencies of a single particle upon hydration (i.e., RH=85% for this study) at a specific visible wavelength (e.g.,  $\lambda$  = 525 nm) could be simulated using the Mie model. It should be noted that the aerosol diameter growth factor, g(RH), was necessary for calculating aerosol optical properties due to the size growth after humidification (Brock et al., 2016; Tan et al.,

2024). g(RH) is normally determined by the aerosol hygroscopicity parameter  $\kappa$ , and the bulk aerosol  $\kappa_{f(RH)}$  of this study was derived from the f(RH) measurements based on the method proposed by Kuang et al. (2017). According to the size distributions (Fig.S5) measured during the study period, the aerosol population was typically divided into the ultrafine ( $D_p < 100$  nm; Uf.) and accumulation ( $D_p \ge 100$  nm; Acc.) modes. Although the size-resolved  $\kappa$  results were unavailable, the mean  $\kappa_i$  for both ultrafine and accumulation mode particles could be roughly estimated assuming that  $\kappa_{f(RH)}$  is a linear combination of volume-weighted  $\kappa_i$  for different modes (e.g.,  $\kappa_{f(RH)}$ = VFUf.\* $\kappa_{Uf.}$ + VFAcc.\* $\kappa_{Acc.}$ ; Hong et al., 2024). Given a generally weaker hygroscopicity for the ultrafine mode (Chen et al., 2012; Petters and Kreidenweis, 2007), the mean  $\kappa$  of ultrafine particles was defined to be half of the measured bulk  $\kappa_{f(RH)}$ , and  $\kappa$  of accumulation mode particles can be derived from the bulk  $\kappa_{f(RH)}$  with the corresponding volume fractions (i.e., VFUf. and VFAcc.); consequently, the corresponding g(RH) for both Uf. and Acc. modes can be calculated with the  $\kappa$ -Köhler theory.

The complex refractive index is another critical input parameter for the Mie model, with the real part of complex refractive index (n) determining the aerosol light scattering ability. Under the assumption of a fixed n for dry aerosols ( $n_{\rm dry} = 1.53$ ) in this study, the volume-weighted n of hydrated particles can be derived with  $n_{\rm dry}$  and f(RH)-derived volume fractions of uptake water,  $f_W$  (Chen et al., 2012):  $n = 1.33 * f_W + 1.53 * (1-f_W)$ , where 1.33 is the n of pure water (Jung et al., 2016). Hence, the efficiencies of  $\sigma_{\rm sca}$ ,  $\sigma_{\rm bsca}$  and HBF after hygroscopic growth could be simulated with the time-averaged dry PNSD, the mean g(RH) of Uf. Mode (1.15) and Acc. mode (1.27), and the mean n of humidified aerosols (1.44) for the observation period. The theoretically calculated results were displayed in Figure R7.

Different from the significant ascending trend of  $\sigma_{sca}$  efficiencies within the size range of around 100~500 nm (Fig. R7a), the efficiency of  $\sigma_{bsca}$  (thereby HBF) presents a rather more periodic fluctuation with the particle size. This further leads to the pronounced variations in efficiencies of both  $\sigma_{bsca}$  and HBF upon hydration (Fig. R7b), and the enhancement of HBF525, RH efficiency exhibited distinct size-dependent

sensitivity regimes. Namely, the efficiency ratio of HBF525, RH/HBF525 remained approximately 1 when  $D_p$ <100 nm, and the ratio gradually decreased below 1 with the particle size (in the range of 100~250 nm); however, it increased abruptly and largely exceeded 1 when  $D_P$  >250 nm, kind of with an oscillatory fluctuation. For instance, the efficiency of HBF525, RH/HBF525 could be above 24 (even reaching 30) when  $D_P$  was 289 nm (445 nm). Such a high level of HBF525, RH/HBF525 was actually observed in our study, with the sampled aerosol population contributed predominantly by particles smaller than 500 nm.

Figure R7. Size-dependent efficiencies of (a) light scattering (the black line), backscattering (the red line) and HBF (the blue line) in dry conditions, as well as (b) the enhancements in corresponding efficiencies of light scattering (the black line), backscattering (the red line) and HBF (the blue line) at  $\lambda = 525$  nm simulated with the Mie theory.

To further investigate influences of the aerosol size distribution, hygroscopic properties, and composition-dependent complex refractive index on the HBF525, RH/HBF525 ratio, we conducted a sensitivity analysis with the measured data specifically for both P1 and P2 NPF days using the Mie model. Previous studies commonly assumed aerosol number size distributions as a combination of multi-lognormal distribution functions, with each mode representing a distinct particle population (Hussein et al., 2004):

$$\frac{dN}{d\log D_{P}} = \sum_{i=1}^{n} \frac{N_{t,i}}{\sqrt{2\pi} \log \sigma_{g,i}} \exp \left[ -\frac{(\log D_{P} - \log \overline{D_{Pg,i}})^{2}}{2 \log^{2} \sigma_{g,i}} \right]$$
(2)

Where the three representative parameters, i.e., the total number concentration  $N_{t,i}$ , the geometric standard deviation (GSD)  $\sigma_{g,i}$ , and the geometrical mean diameter  $D_{Pg,i}$ , can be used to characterize an individual mode i; and n is the number of individual modes (Hussein et al., 2004). In this study, the measured PNSD data on NPF days during P1 and P2 periods were normally fitted into two modes: the predominant Uf. mode and the other one dominated by Acc. Mode particles. As a result, nine parameters were employed in the Mie model: four parameter pairs ( $D_{Pg}$ , GSD,  $N_t$  and g(RH)) for both Uf. and Acc. mode particles, along with the mean n of the bulk aerosol population upon hydration. Further, the HBF525, RH/HBF525 can be simplified as a function of aerosol size distribution (i.e.,  $D_{Pg}$ , GSD,  $N_t$ ), water uptake (e.g., g(RH)), and n as below:

HBF 525, RH/HBF 525 =
$$f(D_{Pg}, GSD, N_{t}, g(RH), n)$$
 (3)

The influence of a specific parameter on the HBF525, RH/HBF525 was evaluated by fixing all the other parameters at their measured mean values and computing HBF525, RH/HBF525 ratios across the range of this target parameter. For instance, the sensitivity of DPg could be illustrated as:

$$f(HBF_{525, RH}/HBF_{525}, D_{Pg}) = f(D_{Pg}, \overline{GSD}, \overline{N_t}, \overline{g(RH)}, \overline{n})$$
(4)

The measured mean value and variation range of each parameter were summarized in Table R1. The ranges of  $D_{Pg}$ , GSD,  $N_t$  and g(RH) were determined based on field measurements of this study. Zhao et al. (2021) reported that n of diverse aerosol populations could range from 1.36 to 1.78 across different Chinese cities, and this study constrained n to vary from 1.3 (nearly pure water of 1.33; Jung et al., 2016) to 1.8 (similar to black carbon of approximately 1.87; Schkolnik et al., 2007) in the modeling framework.

Table R1. A summary of the input parameters for the sensitivity analysis with the Mie models.

|                            | Variable      | Mode | Mean | Range   |
|----------------------------|---------------|------|------|---------|
| P1 NPF polluted | $D_{Pg}$ (nm) | Uf.  | 39   | 14-100  |
|                            |               | Acc. | 173  | 100-300 |
|                            | GSD           | Uf.  | 1.69 | 1.2-2.1 |

|                  |                             | Acc. | 1.56   | 1.2-2.7      |
|------------------|-----------------------------|------|--------|--------------|
|                  | M (43)                      | Uf.  | 16,844 | 2,000-28,000 |
|                  | $N_t$ (#·cm -3 ) | Acc. | 2,311  | 1,000-5,500  |
|                  | g(RH)                       | Uf.  | 1.14   | 1.0-1.3      |
|                  |                             | Acc. | 1.26   | 1.0-1.3      |
|                  | n                           | /    | 1.45   | 1.3-1.8      |
| P2 NPF clean, HW | $D_{Pg}$ (nm)               | Uf.  | 39     | 14-100       |
|                  |                             | Acc. | 150    | 100-300      |
|                  | GSD                         | Uf.  | 1.46   | 1.2-2.1      |
|                  |                             | Acc. | 1.65   | 1.2-2.7      |
|                  | $N_t$ (#-cm-3)              | Uf.  | 14,963 | 2,000-28,000 |
|                  |                             | Acc. | 2,251  | 1,000-5,500  |
|                  | g(RH)                       | Uf.  | 1.15   | 1.0-1.3      |
|                  |                             | Acc. | 1.27   | 1.0-1.3      |
|                  | n                           | /    | 1.44   | 1.3-1.8      |

The sensitivity tests revealed that the calculated HBF525, RH/HBF525 ratios commonly exceeded 1, with enhanced values for P2 NPFclean, HW days (Figures R8-9), which is consistent with the observations. The influences of the PNSD and hygroscopicity related parameters (i.e., DPg, GSD, Nt and g(RH)) on HBF525, RH/HBF525 were displayed in Figure R8. Generally, the pattern of the PNSD (e.g., DPg and GSD) exhibited a much more significant impact on the HBF525, RH/HBF525 ratio than that of the total particle number concentration (Nt), the influence of which was almost negligible as indicated by the insignificant changes in the calculated HBF525, RH/HBF525 ratios. Particularly, the HBF525, RH/HBF525 ratio increased evidently with the mode diameter and GSD (e.g., HBF525, RH/HBF525 >1.5 at DPg = 200~300 nm and GSD >1.8) of the Acc. mode particles, specifically for smaller DPg and GSD (i.e., narrower distributions) of Uf. mode particles during NPF days (Figure R8a1-b2). However, HBF525, RH/HBF525 tends to be less sensitive to g(RH)

regardless of the aerosol mode, likely due to the predefined mode diameters and the smaller variation range of g(RH) used for the sensitivity tests (Table R1). Furthermore, HBF525, RH/HBF525 increased notably with n of hydrated particles, with a pronounced enhancement when n exceeded a threshold value (e.g., ~1.5) especially during P2 NPF days (Figure R8). The joint impacts of aerosol hygroscopic growth on HBF525, RH/HBF525 would depend on both the shift in the distribution pattern with the size growth and simultaneously lowered n due to enhanced liquid water content, highlighting the complex interactions of aerosol size distribution, water uptake, and optical properties in ambient atmospheric environment.

Figure R8. The relationships between the  $HBF_{525, RH}/HBF_{525}$  ratios and the  $D_{pg}$  (a), GSD (b),  $N_t$  (c), g(RH) (d) of two modes particleas. The left (right) column was corresponding to the P1  $NPF_{polluted}$  (P2  $NPF_{clean, HW}$ ) days.

Figure R9. The variations of the  $HBF_{525, RH}/HBF_{525}$  ratios with n levels on the different NPF days.

Another possible explanation could be related to the non-spherical particle morphology upon hygroscopic growth, which was hardly considered due to the precondition of spherical particles in the Mie model. Heterogeneous mixing of chemical components (e.g., secondary inorganics, organic coatings, as well as black carbon) can lead to the heterogeneity in particle bulk hygroscopicity (Yuan and Zhao, 2023), which may further affect the morphology of humidified particles. The more hydrophilic compositions would tend to be more spherical, otherwise with irregular shapes (Giordano et al., 2015; Tan et al., 2020; Tritscher et al., 2011). Furthermore, potential heterogeneity in the humidification process across the "wet" nephelometer chamber may have caused particles to partly exist in a semi-solid phase state (Tang et al., 2019). This combination of heterogeneity in aerosol hygroscopicity and hydration processes likely contributed to the non-spherical particle morphology even after humidification.

In summary, the observed elevated HBF525, RH/HBF525 ratios in this study can be attributed to the following aspects. Firstly, it could be related to the shift of aerosol size distribution toward larger accumulation-mode sizes (e.g., more sensitive to the enhancement in HBF) during the subsequent growth of both pre-existing and newly formed particles or upon hydration in the "wet" nephelometer. Additionally, the dominance of organic components during the study period likely introduced

heterogeneity in aerosol hygroscopicity, which may alter particle morphology upon water uptake. Moreover, aerosol n tends to increase with the aging process of organic components (Moise et al., 2015; Zhao et al., 2021), suggesting that particles may have higher n values during P2 period with the higher photooxidation intensity, thereby contributing to the observed enhancement in the HBF525, RH/HBF525 ratio.

We have updated the manuscript as follows:

L582-591: Additionally, the predominant organic components when heterogeneously mixed with diverse chemical compositions (e.g., inorganics and black carbon) likely introduced the heterogeneity in aerosol hygroscopicity (Yuan and Zhao, 2023), which may alter particle morphology thereby optical properties upon water uptake (Giordano et al., 2015; Tan et al., 2020; Tritscher et al., 2011). The efficient evaporation of organic coatings under extremely hot conditions could also contribute to the change in particle morphology (e.g., non-spherical inregular shapes) upon humidification, as evidenced by a recent study that high temperature conditions could accelerate the evaporation rate of SOA (Li et al., 2019).

L599-621: Another possible reason is the distinct size dependences of both light scattering and backscattering efficiencies (Fig. S11a), with much more significant enhancements in the backscattering efficiency thereby HBF specifically of accumulation mode particles after hygroscopic growth (Fig. S11b). As reflected by the Mie model, although the abundant newly formed particles were generally optically-insensitive (e.g., below 100 nm), their contributions to σsca, 525 and especially to σbsca, 525 could be amplified upon humidification (Fig. S11b). Besides, the shift of size distribution towards larger accumulation-mode particles could also result in a significant elevation in HBF525, RH/HBF525 ratios, especially under the condition of a smaller mode diameter and narrower distribution of ultrafine-mode particles (e.g., during NPF events) (Fig. S15a1-b2 for the theoretical sensitivity tests of Sect. S9 in the supplement). Furthermore, the HBF525, RH/HBF525 ratio exhibited a significant positive correlation with the real part of complex refractive index (n) of bulk aerosols (Fig. S16), and n tends to increase with the aging process of organic species (Moise et al., 2015; Zhao et al., 2021). In this sense, the evolution of both aerosol size

distribution pattern and chemical compositions, combined with the heterogeneity in aerosol hygroscopicty, could potentially change particle morphology and optical properties (e.g., complex refractive index and elevated HBF525, RH) particularly during heatwave-influenced NPFclean, HW days, characterized with the smallest aerosol  $R_{\rm eff}$  (102.8  $\pm$  12.4 nm) (Figure. S6), lowest number concentration (1897.0  $\pm$  680.8 cm-3) and fraction (0.20  $\pm$  0.10) of accumulation mode particles, intensified photooxidation, and a higher SOC/OC ratio.

**Updates in the reference list:**

Giordano, M., Espinoza, C., and Asa-Awuku, A.: Experimentally measured morphology of biomass burning aerosol and its impacts on CCN ability, Atmos. Chem. Phys., 15, 1807–1821, https://doi.org/10.5194/acp-15-1807-2015, 2015.

Moise, T., Flores, J. M., and Rudich, Y.: Optical Properties of Secondary Organic Aerosols and Their Changes by Chemical Processes, Chem. Rev., 115, 4400–4439, https://doi.org/10.1021/cr5005259, 2015.

Tritscher, T., Jurnyi, Z., Martin, M., Chirico, R., Gysel, M., Heringa, M. F., Decarlo, P. F., Sierau, B., Prévt, A. S. H., Weingartner, E., and Baltensperger, U.: Changes of hygroscopicity and morphology during ageing of diesel soot, Environ. Res. Lett., 6, https://doi.org/10.1088/1748-9326/6/3/034026, 2011.

Yuan, L. and Zhao, C.: Quantifying particle-To-particle heterogeneity in aerosol hygroscopicity, Atmos. Chem. Phys., 23, 3195–3205, https://doi.org/10.5194/acp-23-3195-2023, 2023.

Zhao, G., Hu, M., Fang, X., Tan, T., Xiao, Y., Du, Z., Zheng, J., Shang, D., Wu, Z., Guo, S., and Zhao, C.: Larger than expected variation range in the real part of the refractive index for ambient aerosols in China, Sci. Total Environ., 779, 146443, https://doi.org/10.1016/j.scitotenv.2021.146443, 2021.

We have added the detailed Mie scattering simulations in S7 and replaced the original Figure S11a into the Figure R7, the sensitivity analysis and results have been added in S9, Figures S15-16 and Table S3 in Supplement.

**Updates in the reference list of Supplement:**

Brock, C. A., Wagner, N. L., Anderson, B. E., Attwood, A. R., Beyersdorf, A., Campuzano-Jost, P., Carlton, A. G., Day, D. A., Diskin, G. S., Gordon, T. D., Jimenez, J. L., Lack, D. A., Liao, J., Markovic, M. Z., Middlebrook, A. M., Ng, N. L., Perring, A. E., Richardson, M. S., Schwarz, J. P., Washenfelder, R. A., Welti, A., Xu, L., Ziemba, L. D., and Murphy, D. M.: Aerosol optical properties in the southeastern United States in summer - Part 1: Hygroscopic growth, Atmos. Chem. Phys., 16, 4987–5007, https://doi.org/10.5194/acp-16-4987-2016, 2016.

Chen, J., Zhao, C. S., Ma, N., Liu, P. F., Göbel, T., Hallbauer, E., Deng, Z. Z., Ran, L., Xu, W. Y., Liang, Z., Liu, H. J., Yan, P., Zhou, X. J., and Wiedensohler, A.: A parameterization of low visibilities for hazy days in the North China Plain, Atmos. Chem. Phys., 12, 4935–4950, https://doi.org/10.5194/acp-12-4935-2012, 2012.

Hong, J., Ma, J., Ma, N., Shi, J., Xu, W., Zhang, G., Zhu, S., Zhang, S., Tang, M., Pan, X., Xie, L., Li, G., Kuhn, U., Yan, C., Qi, X., Zha, Q., Nie, W., Tao, J., He, Y., Zhou, Y., Sun, Y., Xu, H., Liu, L., Cai, R., Zhou, G., Kuang, Y., Yuan, B., Wang, Q., Petäjä, T., Kerminen, V. M., Kulmala, M., Cheng, Y., and Su, H.: Low Hygroscopicity of Newly Formed Particles on the North China Plain and Its Implications for Nanoparticle Growth, Geophys. Res. Lett., 51, https://doi.org/10.1029/2023GL107516, 2024.

Hussein, T., Puustinen, A., Aalto, P. P., Mäkelä, J. M., Hämeri, K., and Kulmala, M.: Urban aerosol number size distributions, Atmos. Chem. Phys., 4, 391–411, https://doi.org/10.5194/acp-4-391-2004, 2004.

Jung, C. H., Shin, H. J., Lee, J. Y., and Kim, Y. P.: Sensitivity and contribution of organic aerosols to aerosol optical properties based on their refractive index and hygroscopicity, Atmosphere (Basel)., 7, https://doi.org/10.3390/atmos7050065, 2016.

Moise, T., Flores, J. M., and Rudich, Y.: Optical Properties of Secondary Organic Aerosols and Their Changes by Chemical Processes, Chem. Rev., 115, 4400–4439, https://doi.org/10.1021/cr5005259, 2015.

Schkolnik, G., Chand, D., Hoffer, A., Andreae, M. O., Erlick, C., Swietlicki, E., and Rudich, Y.: Constraining the density and complex refractive index of elemental and organic carbon in biomass burning aerosol using optical and chemical measurements, Atmos. Environ., 41, 1107–1118, https://doi.org/10.1016/j.atmosenv.2006.09.035, 2007.

Tan, F., Zhang, H., Xia, K., Jing, B., Li, X., Tong, S., and Ge, M.: Hygroscopic behavior and aerosol chemistry of atmospheric particles containing organic acids and inorganic salts, npj Clim. Atmos. Sci., 7, 1–21, https://doi.org/10.1038/s41612-024-00752-9, 2024.

Zhao, G., Hu, M., Fang, X., Tan, T., Xiao, Y., Du, Z., Zheng, J., Shang, D., Wu, Z., Guo, S., and Zhao, C.: Larger than expected variation range in the real part of the refractive index for ambient aerosols in China, Sci. Total Environ., 779, 146443, https://doi.org/10.1016/j.scitotenv.2021.146443, 2021.

**References**

- Carrico, C. M., Kus, P., Rood, M. J., Quinn, P. K., & Bates, T. S. (2003). Mixtures of pollution, dust, sea salt, and volcanic aerosol during ACE-Asia: Radiative properties as a function of relative humidity. *Journal of Geophysical Research:*Atmospheres, 108(D23). https://doi.org/10.1029/2003jd003405
- Fierz-Schmidhauser, R., Zieger, P., Gysel, M., Kammermann, L., DeCarlo, P. F., Baltensperger, U., & Weingartner, E. (2010). Measured and predicted aerosol light scattering enhancement factors at the high alpine site Jungfraujoch. *Atmospheric Chemistry and Physics*, 10(5), 2319–2333. https://doi.org/10.5194/acp-10-2319-2010

---

## Author Response (AR3)

Dear Editor,

We thank for all the constructive comments and suggestions from referees. We have carefully addressed and provided detailed explanations for their concerns. Point-by-point responses to the suggestions, corresponding updates with the revised manuscript, and the finalized version have been uploaded.

In the following, original suggestions, our response, and updates on the revised manuscript are shown in **bold**, normal, and *italic*, respectively.

Kind Regards,

Jing Chen, Yuhang Hao, and Peizhao Li

**Anonymous Referee #2**

**General comments:**

I thank the authors for their efforts and for satisfactorily addressing my earlier comments and suggestions. Although I have indicated the need for minor revisions, the comment below may be considered a major one and must be addressed before the manuscript is accepted.

**Response:** We appreciate the reviewer for the supportive comments.

**Specific comments:**

RC1. Regarding Figure R4 in general (and specifically R4a3 and R4b3): Did the authors use the entire time period or only the NPF event time period (8 am -10 pm LT) to create the box-whisker plots? This appears to contradict Figures 4a and 4b, where NFaccu and VFaccu are clearly higher on non-event days than on NPF event days (for both P1 and P2). Does this suggest that NPF does not contribute to the accumulation mode (both number and volume) during either time period?

**Response:** We use the entire days' data (24-h) for the box-whisker plots of corresponding concentrations in Figure R4 in our previous response, while Figures 4a-b were the time-averaged diurnal variation results of both number and volume fractions for the accumulation mode particles. To avoid unnecessary misleading, the caption of Figure S7 has been adjusted into "The number concentrations (left column: a1-a3) and volume concentrations (right column: b1-b3) of different mode particles for the corresponding NPF and non-event days during both P1 and P2 periods.".

Accumulation mode particles are suggested to originate predominantly from direct emissions or aging of pre-existing particles (e.g., local sources, regional transport), with NPF contributing marginally to the total number and volume concentrations of the accumulation mode (i.e., without significant increases during the event time periods as shown in Figure S8). Nevertheless, NPF events could significantly modulate variations in NFAcc. and VFAcc. (Figure 4a, b), mainly due to the

shift in the particle number size distribution driven by distinctly enhanced abundances of both nucleation and Aitken mode particles during NPF events. For instance, the discrepancies of NFAcc. and VFAcc. during the non-NPF time window (i.e., unshaded areas in Figures 4a- b) were different from that for the NPF event period in the P1 period. Namely, a higher level of NFAcc. (VFAcc.) was observed during the non-NPF time window on the P1 NPFpolluted days than on non-event days, yet both NFAcc. and VFAcc. decreased substantially upon NPF event onset. This not only signifies the importance of local accumulation and aging processes on the accumulation mode particles during non-event time window in relatively polluted P1 period, but also highlights the crucial role of NPF in regulating the size distribution pattern thereby the relative abundance of the accumulation mode particles during NPF events. The below Figure R1 shows the mean NFAcc. and VFAcc. results of NPFpolluted days (the entire days) and NPFpolluted event time periods (08:00-22:00 LT) during the P1 period, as well as that for the corresponding non-event cases. The mean NFAcc. and VFAcc. were quite similar on the whole NPFpolluted and non-event days, whereas significantly lower (specifically for NFAcc.) during NPFpolluted event time periods than that either for the same period of non-event cases or the mean levels for whole days' data (Figure R1). It is noted that the NFAcc. and VFAcc. levels were even lower on the NPFclean, HW days than non-event days during P2 period (Figures 4a-b), possibly due to the rather cleaner environment which amplified the corresponding impacts of NPF on lowering the relative abundances of the accumulation mode.

Updated on the main text:

**L499-503:** This is mainly due to the explosive formation of ultrafine particles and subsequent growth on NPF days, significantly altering aerosol size distributions and inducing large fluctuations in the NFAcc. and VFAcc. in comparison to that on non-event days, especially during the P2 period (as shaded in Fig. 4a-b).

**Figure R1.** Box plots of  $NF_{Acc.}$  (a) and  $VF_{Acc.}$  (b) for both the whole days and just during the 08:00-22:00 LT time window in P1 period.

RC2. Additionally, the smallest aerosol effective radius is ~80 nm. What fraction of newly formed particles grows to this size and subsequently to the accumulation mode (during both P1 and P2), such that they can be associated with changes in aerosol hygroscopicity resulting from new particle formation?

Response: We're afraid that the newly formed particles could rarely grow into the accumulation mode (e.g.,  $D_p > 100$  nm) under dry conditions (Kerminen et al., 2018). As shown in Figure S6, the SMPS-determined mean  $R_{eff}$  decreased to around 60 nm (50 nm) after the occurrence of the P1 NPFpolluted (P2 NPFclean, HW) event. Figure R1 has further demonstrated decreases rather than increases in both NFAcc. and VFAcc. during NPF events, although the P1 NPFpolluted events exhibited a relatively larger level of  $R_{eff}$ . Based on the growth rate (GR) and the duration of NPF events obtained from our observations, the maximum attainable sizes for newly formed particles during both periods can be accordingly estimated. Specifically, the newly formed particles can grow into a maximum dry size of 74.1  $\pm$  17.0 nm during P1 NPFpolluted events, while they can grow to 81.5  $\pm$  10.3 nm due to prolonged subsequent growth till the evening during P2, despite of a relatively lower GR. Unless with a pronounced size growth upon aging or hydration (e.g.,  $D_p > 200$  nm), these maximally grown dry particles would not significantly influence the total light scattering by themselves, as evidenced by the size-resolved light scattering results in Figure S13. Nevertheless,

variations in optical and hygroscopic properties of the aerosol population due to the shift in aerosol size distributions (e.g., altered fractions of ultrafine and accumulation modes induced by NPF events along with the hygroscopic growth and aging processes) could be distinct and nonnegligible, as reflected in Figure S11a-b and Figure S16. In summary, it could be difficult to affect the aerosol optical hygroscopicity solely by the subsequent growth of newly formed particles due to insignificant light scattering efficiencies in the dry condition, yet the hydrated and aged new particles may modulate the aerosol optical and hygroscopic properties largely under specific alterations in the aerosol size distribution patterns.

**Anonymous Referee #4**

While I do think that this manuscript presents interesting data I also believe that the comments of both referees are well substantiated. My own impression very much conforms to referee 2 which might have major implications on the conclusions drawn.

**Response:** Thanks for the constructive comments and suggestions.

Specifically the missing nucleation mode particles during the heatwave indicate advection of newly formed particles rather than in-situ formation. This has been addressed in the revised manuscript for horizontal transport but does not take into account potential down mixing during boundary layer evolution. In any case, my concern is that whatever the reason for the lacking newly formed particles is one could equally argue in the other direction that extremely hot weather conditions could suppress NPF at this location. Under the conditions shown, formation and growth rates will simply not be comparable to other periods and will lead to wrong results/interpretation as the sudden appearance is not associated with growth and formation but with changing air mass. I think that the conclusions should be amended accordingly. Maybe also the title should reflect that these data were taken "...during a heat wave..." Furthermore, the text several times mentions "...during weather extremes (e.g., heat waves)..." which does imply also weather extremes beyond heat waves and is not justified in my opinion. I'm therefore ambivalent about this manuscript. I think it is worth publishing but it may still need some optimization in the wording and careful interpretation.

**Response:** Thanks for the comments. We acknowledge the potential impact of the vertical transport on NPF events (Crumeyrolle et al., 2010; Lai et al., 2022a; Platis et al., 2016). Actually, the calculated cumulative contributions of transportation (see Figure R1 in our latest response) included both horizontal and vertical transport. As shown in eq. (1-2) and Figure 1 of Cai et al. (2018):

$$\frac{dN_{[i,j]}}{dt} = GR_{i}n_{i} - GR_{j}n_{j} + CoagSrc_{[i,j]} - CoagSnk_{[i,j]} + TR_{[i,j]}$$
(1)

where the subscripts i and j correspond to the specific particle diameters di and dj, respectively;  $N_{[i,j]}$  is the number concentration of particles ranging from di and dj; GR is the condensational growth rate;  $CoagSrc_{[i,j]}$  and  $CoagSnk_{[i,j]}$  are the formation and loss rates due to coagulation (Cai et al., 2018).  $TR_{[i,j]}$  is the transport term:

$$TR_{[i,j]} = \lim_{dt \to 0} \frac{N_{[i,j]}(t + dt, z^2) - N_{[i,j]}(t + dt, z^1)}{dt}$$
(2)

here, z corresponds to the aerosol populations from different air masses (i.e., either horizontally or vertically) (Cai et al., 2018). A higher mixing layer height (MLH) was indeed observed under the P2 heatwaves, likely favorable for the vertical mixing. Figure R2 displayed the diurnal variations of MLH and the contribution of transport to the nucleation-mode particle number concentration on a NPFclean, HW day. Specifically, MLH gradually developed from 8:00 LT to 17:00 LT and then sharply decreased, with a relatively lower level at night. However, the cumulative contributions of transport significantly increased from 9:00 LT to 11:00 LT and maintained increasing slowly till 21:00 LT, without any observed weakening accompanied with the sharp decline in MLH. In this sense, the contribution of vertical transport (as partly reflected by the MLH evolution) may not be so significant as the horizontal transport to NPF in this study. Considering that our data only represented NPF events during a heatwave of 2022 in southwest China, the findings may not fully capture the impacts of vertical mixing on NPF under different heatwaves. We therefore recommend future investigations to systematically evaluate both horizontal and vertical transport influences on NPF events, especially during heatwaves in diverse regions.

**Figure R2.** Diurnal evolutions of MLH and the cumulative contribution of transport to the nucleation mode particle number concentration on a  $NPF_{clean, HW}$  day (i.e., 9 August 2022 as an example).

To further investigate the formation and subsequent growth of nucleation mode particles under heatwaves, we plotted the geometric mean diameter of fitted ultrafine mode (GMDUf, see eq.5 in S9 of the supplement) from NPF onset until 19:00 LT using multi-lognormal distribution functions. Unlike the commonly observed discontinuous variations in GMDUf largely driven by vertical transport during NPF events (Lai et al., 2022; Wu et al., 2024), the nearly uninterrupted GMDUf results derived from our size distribution data (Figure R3) could clearly demonstrate the local formation of new particles (<25 nm) and the subsequent complete growth process. This highlights that the increase in NNuc was not entirely due to the transport, at least not so significantly affected by vertical mixing under heatwaves. The FR and GR<25 nm remain comparable across different periods of our field observations, particularly when the wind direction and air mass trajectories were similar (Figure 1d, Figure S4). Additionally, previous studies have also compared the FR and GR of different NPF events dominated by local nucleation and transport, or originated from different air

masses (Chandra et al., 2016; Hussein et al., 2009; Komppula et al., 2006; Kulmala et al., 2004; Shang et al., 2023). For example, Chandra et al. (2016) observed different NPF events starting from 5 nm and from~10 nm due to upstream transport in Fukue Island. Komppula et al. (2006) found a relatively good correlation in GR of two NPF cases in air mass transported between two measurement sites in Northern Finland. Although the "banana-shape" NPF event did not start from the minimum size of SMPS during the P2 period, the evidently continuous GMDUf can support that the heatwaves did not remarkably suppress NPF events (Figure R3). As both P1 and P2 periods exhibited local formation followed by subsequent growth of NPF, the comparative analysis of FR, GR, and other aerosol physicochemical properties during different NPF events can be regarded as methodologically justified.

Figure R3. Overview of the measured PNSD and geometrical mean diameter of the fitted ultrafine mode (GMD $_{Uf}$  black dots) during the P2 NPF $_{clean, HW}$  days.

The comparable NPF occurrence frequencies during P2 (50.0%), P1 (53.8%), and P22023 (53.8%; without heatwaves in summer 2023) further indicated that NPF events were not likely suppressed markedly by heatwaves in summer 2022. However, we have to underline that heatwaves did have a notable influence on NPFclean, HW events, as reflected in the potential contributions of transport, the earlier occurrence time, as well as observed reductions in FR, GR, and Reff. Given the more and more frequent heatwave events with the changing climate, we primarily focus on the significant changes observed in both NPF events and aerosol optical and hygroscopic properties against the background of heatwaves.

To avoid controversial conclusions about the impacts of heatwaves on NPF (e.g., either favor or suppress), here we chose to objectively describe the differences in statistical characteristics of NPF events observed under heatwaves, while the detailed formation mechanisms of different NPF events are out of the scope of the current study and will be explored in future work. Although it is difficult to quantify the individual contribution of vertical transport in this study, we have added corresponding discussion on the potential contribution of transport to the main text and amended the Conclusions accordingly. Figure R3 has been added into the revised Supplement (i.e., Figure S15):

L358-367: Different from that of the P1 NPFpolluted cases, the P2 NPFclean, HW event did not start from the minimum size, and the reduced NNuc. during P2 period was likely attributed to the influence of transport on the local nucleation and growth process (Fig. S4; Cai et al., 2023; Lee et al., 2019). Namely, some nucleation mode particles transported from upwind regions or from the mixing layer downwards had undergone atmospheric aging thereby a certain degree of growth upon arrival (Cai et al., 2023; Lai et al., 2022; Platis et al., 2016), resulting in relatively lower concentrations of smaller-sized particles than the case of locally formed. However, the local formation of sub-25 nm particles and the continuous growth process were still distinctly observed under heatwaves (Fig. 1i, Figs. S6, S15).

**Updates in the Conclusions and implications:**

**L682-685:** Although the air masses and the occurrence frequencies of NPF events were similar during different periods, NPF events exhibited distinct characteristics during the normally hot (P1, relatively polluted) and heatwaves-dominated (P2, quite clean) periods.

**L686-690:** NPFclean, HW events that occurred during the heatwave P2 period were observed with lower CS, CoagS, FR and GR, as well as smaller  $R_{eff}$  and  $D_{mode}$ , than P1 NPFpolluted cases. According to the measured PNSDs, the P1 NPFpolluted events were mainly driven by local growth, while NPFclean, HW events may be largely affected by transport under heatwaves.

L710-713: This was likely due to the observed lower FR and GR caused by possible evaporation of both unstable clusters and particle coatings under heatwaves (Bousiotis et al., 2021; Cusack et al., 2013; Deng et al., 2020; Garmash et al., 2024), L734-739: This study revealed divergent changes in aerosol optical and hygroscopic properties on different NPF days, thereby modulating the aerosol radiative forcing distinctly during a heatwave in summer 2022. A comprehensive understanding of the formation mechanisms of different NPF events (e.g., local formation versus the horizontal or vertical transport) in diverse environment is crucial in the future.

**Updates in the reference list:**

Lai, S., Hai, S., Gao, Y., Wang, Y., Sheng, L., Lupascu, A., Ding, A., Nie, W., Qi, X., Huang, X., Chi, X., Zhao, C., Zhao, B., Shrivastava, M., Fast, J. D., Yao, X., and Gao, H.: The striking effect of vertical mixing in the planetary boundary layer on new particle formation in the Yangtze River Delta, Sci. Total Environ., 829, 154607, https://doi.org/10.1016/j.scitotenv.2022.154607, 2022.

Platis, A., Altstädter, B., Wehner, B., Wildmann, N., Lampert, A., Hermann, M., Birmili, W., and Bange, J.: An Observational Case Study on the Influence of Atmospheric Boundary-Layer Dynamics on New Particle Formation, Boundary-Layer Meteorol., 158, 67–92, https://doi.org/10.1007/s10546-015-0084-y, 2016.

We have adjusted the title into "Divergent changes in aerosol optical hygroscopicity and new particle formation during a heatwave of summer 2022", and all the ambiguous statements of 'weather extremes (e.g., heatwaves)' have also been revised into 'heatwaves'.

L43-45: Further in-depth exploration on molecular-level characterizations and aerosol radiative impacts of both direct and indirect interactions under heatwaves and other weather extremes with the warming climate are recommended.

*L135-136*: specifically under heatwaves with the changing climate.

**L667-668:** This highlights the needs for further in-depth exploration on aerosol radiative impacts under heatwaves with the changing climate,

L739-744: The last but not the least, further explorations on detailed molecular-scale characterizations (e.g., molecular structures and compositions of newly and secondary formed particles, as well as particle morphology) and aerosol radiative impacts including the aerosol-cloud interactions of both heatwaves and other weather extremes with the changing climate are highly recommended.

**References**

Chandra, I., Kim, S., Seto, T., Otani, Y., Takami, A., Yoshino, A., Irei, S., Park, K., Takamura, T., Kaneyasu, N., and Hatakeyama, S.: New particle formation under the influence of the long-range transport of air pollutants in East Asia, Atmos. Environ., 141, 30–40, https://doi.org/10.1016/j.atmosenv.2016.06.040, 2016.

Crumeyrolle, S., Manninen, H. E., Sellegri, K., Roberts, G., Gomes, L., Kulmala, M., Weigel, R., Laj, P., and Schwarzenboeck, A.: New particle formation events measured on board the ATR-42 aircraft during the EUCAARI campaign, Atmos. Chem. Phys., 10, 6721–6735, https://doi.org/10.5194/acp-10-6721-2010, 2010.

Hussein, T., Junninen, H., Tunved, P., Kristensson, A., Dal Maso, M., Riipinen, I., Aalto, P. P., Hansson, H. C., Swietlicki, E., and Kulmala, M.: Time span and spatial scale of regional new particle formation events over Finland and Southern Sweden, Atmos. Chem. Phys., 9, 4699–4716, https://doi.org/10.5194/acp-9-4699-2009, 2009.

Komppula, M., Sihto, S. L., Korhonen, H., Lihavainen, H., Kerminen, V. M., Kulmala, M., and Viisanen, Y.: New particle formation in air mass transported between two measurement sites in Northern Finland, Atmos. Chem. Phys., 6, 2811–2824, https://doi.org/10.5194/acp-6-2811-2006, 2006.

Kulmala, M., Vehkamäki, H., Petäjä, T., Dal Maso, M., Lauri, A., Kerminen, V. M., Birmili, W., and McMurry, P. H.: Formation and growth rates of ultrafine atmospheric particles: A review of observations, J. Aerosol Sci., 35, 143–176, https://doi.org/10.1016/j.jaerosci.2003.10.003, 2004.

Wu, H., Li, Z., Hai, S., Gao, Y., Jiang, J., Zhao, B., Cribb, M., Zhang, D., Pu, D., Liu, M., Wang, C., Lan, J., and Wang, Y.: Vertical transport of ultrafine particles and turbulence evolution impact on new particle formation at the surface & Canton Tower, Atmos. Res., 302, 107290, https://doi.org/10.1016/j.atmosres.2024.107290, 2024.

---

## Author Response (AR4)

Dear Editor,

We thank for all the constructive comments and suggestions from the referee. We have carefully addressed and provided detailed explanations for all the concerns. Point-by-point responses to the suggestions, corresponding updates with the revised manuscript, and the finalized version have been uploaded.

In the following, original suggestions, our response, and updates on the revised manuscript are shown in **bold**, normal, and *italic*, respectively.

Kind Regards,

Jing Chen, Yuhang Hao, and Peizhao Li

**Anonymous Referee #2**

RC1. From the title, this study presents divergent changes in aerosol optical hygroscopicity AND new particle formation events, both induced by heatwaves, but conclusions drawn at several places indicate that heatwaves promote NPF, leading to enhanced aerosol hygroscopicity. There is not enough evidence to substantiate this. Note that the aerosol hygroscopicity calculated for total aerosol loading in the atmosphere and NPF's contribution to it is questionable. The manuscript can be accepted, but the authors must tone down statements, including the abstract and conclusion.

**Response:** Thanks for the comments. To avoid unnecessary misleading, we have modified the statements accordingly throughout the main text.

**Updates in the Abstract:**

L25-29: However, mechanisms regulating aerosol optical hygroscopicity during different NPF days, particularly those under heatwaves due to global warming, remain poorly understood. In the 2022 hot summer in urban Chongqing of southwest China, simultaneous measurements of aerosol optical and hygroscopic properties, PNSD, and bulk chemical compositions were conducted.

L35-45: A generally higher f(RH) was observed on NPF days than non-event cases, partly attributed to distinct changes in PNSD patterns during NPF days. Moreover, heatwave-induced stronger photooxidation may intensify the formation of more hygroscopic secondary components and prolong the atmospheric aging/subsequent growth of both pre-existing and newly formed particles, largely contributing to the enhanced f(RH) especially during NPFclean, HW days. The higher f(RH) and lowered Reff could synergistically elevate the aerosol direct radiative forcing, specifically under persistent heatwave conditions. Further in-depth exploration on molecular-level characterizations and aerosol radiative impacts of both direct and indirect interactions under heatwaves with the warming climate are recommended.

**Updates in the Results and discussion:**

L454-460: Given that newly formed particles were too small to significantly impact

the total light scattering (Fig. S11a), this indicates that the atmospheric conditions conducive to the occurrence of NPF may promote further growth (e.g., via intensified/prolonged photooxidation or atmospheric aging processes) of pre-existing particles and newly formed ones, partly contributing to enhanced aerosol optical hygroscopicity as clued from the concurrent variations of ALWC and  $f_W$  in urban Chongqing during hot summer (Asmi et al., 2010; Wang et al., 2019; Wu et al., 2016).

*L516-517:* , probably attributed to the following two aspects.

**L558-562:** Specifically, particles could undergo a longer and more intensified photochemical aging process during  $NPF_{clean, HW}$  days as influenced by persistent heatwaves, which facilitated the secondary formation of hygroscopic aerosols and jointly contributed to a higher f(RH) after 15:00 LT (Fig. 3b).

**L650-652:** Our findings suggest that NPF days may possess a relatively higher aerosol optical hygroscopicity in rather hot environments, e.g., the Basin area and tropical regions.

**L656-659:** Hence, the enhancement of aerosol optical hygroscopicity during the subsequent growth and aging of both pre-existing and newly formed particles possibly exacerbates secondary pollution and even triggers haze events (Hao et al., 2024; Kulmala et al., 2021).

**Updates in the Conclusions and implications:**

**L691-696:** In comparison to the P1 NPFpolluted events, NPFclean, HW occurred approximately one hour earlier and the subsequent growth was longer during P2, likely intensifying the photochemical oxidation and prolonging atmospheric aging processes under heatwaves, thereby modulating the evolution of aerosol size distributions and chemical characteristics differently.

**L704-707:** Specifically, aerosol optical hygroscopicity was observed to be higher during the subsequent growth and aging of both pre-existing particles and newly formed ones on P2 NPFclean, HW days than that for P1 NPFpolluted days,

**L708-709:** Compared with non-event cases, the daily mean f(RH) levels were generally higher on NPF days in the 2022 hot summer of urban Chongqing.

L730-738: The above highlights that heatwaves could influence the NPF

characteristics (e.g., the evolution in the aerosol size distribution pattern and chemical composition) and atmospheric processing (although with a decreased aerosol  $R_{eff}$  and  $D_{mode}$  likely due to evaporation-resulted non-spherical particle morphology under persistently high temperature conditions). Further, variations in the aerosol size distribution and optical hygroscopicity under heatwaves were accompanied with the elevated  $HBF_{525, RH}/HBF_{525}$  ratios, potentially reducing the net solar radiation directly especially in hot summer.

RC2. Authors also stated in their response that "accumulation mode particles are suggested to originate predominantly from direct emissions or aging of pre-existing particles, with NPF contributing marginally to the total number and volume concentrations of the accumulation mode". I do not see a significant enhancement in volume/mass concentrations during NPF events, and also, number concentrations are not very different between NPF and non-events.

Response: Since NPF contributes minimally to the accumulation mode particles, the volume/mass concentrations of the accumulation mode would not enhance significantly during NPF events, instead with only slightly reduced volume fractions of VFAcc. as displayed in the below Figure R1. While in comparison to non-event cases, the explosive formation of new particles commonly leads to a higher number concentration/fraction of both nucleation and Aitken mode particles, coincided with a much lower NFAcc., during NPF event time windows specifically for the heatwave-dominated P2 period with a much cleaner background (i.e., the NPFclean, HW events; Figure R1c, Figure S7). It should be noted that disparities in number concentrations of the accumulation mode are suggested to originate predominantly from variations in pollution levels, emission sources, atmospheric aging degree, etc., rather than the direct contributions of NPF. Hence, the number concentration/fraction of accumulation mode particles was relatively higher on P1 NPFpolluted days than that for non-event cases, largely due to the more polluted environment during P1 NPF periods (Figure R1a, Figure S7, Table S2).

Figure R1. The number and volume fractions of different mode particles for both the whole days (labeled as 'Day') and just during the 08:00-22:00 LT time window (denoted as 'Event') in P1 and P2 period.

**RC3. I suggest removing "other weather extremes" throughout the manuscript; it is vague.**

**Response:** We have removed the corresponding statements throughout the manuscript.